# Electrically switchable van der Waals magnon valves

Guangyi Chen[1,10], Shaomian Qi[1,10], Jianqiao Liu[1], Di Chen[1,2], Jiongjie Wang[3], Shili Yan[2], Yu Zhang[2], Shimin Cao [1,2], Ming Lu[1,2], Shibing Tian[4], Kangyao Chen[1], Peng Yu[5], Zheng Liu [6], X. C. Xie[1,2,7], Jiang Xiao [3], Ryuichi Shindou[1] & Jian-Hao Chen [1,2,8,9✉]

Van der Waals magnets have emerged as a fertile ground for the exploration of highly tunable spin physics and spin-related technology. Two-dimensional (2D) magnons in van der Waals magnets are collective excitation of spins under strong confinement. Although considerable progress has been made in understanding 2D magnons, a crucial magnon device called the van der Waals magnon valve, in which the magnon signal can be completely and repeatedly turned on and off electrically, has yet to be realized. Here we demonstrate such magnon valves based on van der Waals antiferromagnetic insulator MnPS$_3$. By applying DC electric current through the gate electrode, we show that the second harmonic thermal magnon (SHM) signal can be tuned from positive to negative. The guaranteed zero crossing during this tuning demonstrates a complete blocking of SHM transmission, arising from the nonlinear gate dependence of the non-equilibrium magnon density in the 2D spin channel. Using the switchable magnon valves we demonstrate a magnon-based inverter. These results illustrate the potential of van der Waals anti-ferromagnets for studying highly tunable spin-wave physics and for application in magnon-base circuitry in future information technology.

[1] International Center for Quantum Materials, School of Physics, Peking University, Beijing, China. [2] Beijing Academy of Quantum Information Sciences, Beijing, China. [3] Department of Physics and State Key Laboratory of Surface Physics, Fudan University, Shanghai, China. [4] Institute of Physics, Chinese Academy of Sciences, Beijing, China. [5] State Key Laboratory of Optoelectronic Materials and Technologies, School of Materials Science and Engineering, Sun Yat-sen University, Guangzhou, China. [6] School of Materials Science and Engineering, Nanyang Technological University, Singapore, Singapore. [7] CAS Center for Excellence in Topological Quantum Computation, University of Chinese Academy of Sciences, Beijing, China. [8] Key Laboratory for the Physics and Chemistry of Nanodevices, Peking University, Beijing, China. [9] Interdisciplinary Institute of Light-Element Quantum Materials and Research Center for Light-Element Advanced Materials, Peking University, Beijing, China. [10] These authors contributed equally: Guangyi Chen, Shaomian Qi. ✉email: chenjianhao@pku.edu.cn

Van der Waals magnets are recently discovered magnetic materials with covalent bonding within the two-dimensional atomic layers and van der Waals interactions between the layers[1–3]. Owing to the short-range nature of magnetic exchange interaction, van der Waals magnets usually have weak interlayer exchange coupling strengths[4], making the spin system highly two-dimensional and susceptible to external perturbations[5]. Therefore, 2D magnons, quanta of spin waves propagating in van der Waals magnets, are highly tunable collective modes that are of great interest in fundamental science[6–9] and are potentially technologically useful[10–12]. To harness static spin configurations and dynamic spin excitations for potential technological applications, spin valves, and magnon valves are two respective crucial types of devices[13,14]. Recently, spin valves based on van der Waals crystals with a high on–off ratio[15,16] and electrical switchability[17,18] have been demonstrated. The electrically switchable van der Waals magnon valve without varying external magnetic field, however, has yet to be realized.

Owing to the wave nature of magnons, creating "1" in a magnonic circuit (e.g., a state with finite signal) is relatively trivial, while creating "0" (e.g., a state with zero signal) is not. Using two diffusive magnon streams with the same magnitude and opposite directions, it is possible to create "0" at the detector electrode[19]. However, such an operation is closer to signal mixing rather than gating. On the other hand, making use of the concept of "gating" from the charge-based field effect transistor, it has been shown that the magnon conductivity of thin films of three-dimensional (3D) ferrimagnetic insulator yttrium iron garnet could indeed be tuned by passing a current through a strong spin-orbit coupling metal gate electrode[14]. The state of the art of such 3D magnon valve can achieve up to ~13% signal modulation[14], which still falls short in terms of tunability. In this article, we report the realization of van der Waals magnon valves with 100% tunability using thin flakes of van der Waals antiferromagnetic insulator MnPS₃.

## Results

MnPS₃ belongs to a class of layered antiferromagnetic insulators with chemical composition as XPS$_y$ ($X =$ Fe, Cr, Mn, Ni; $y = 3$, 4)[20–22]. It has an energy bandgap of ~3.0 eV for bulk crystals[23] with an easy axis mostly perpendicular to the sample plane[22]. Within each layer, the manganese atoms form a hexagonal structure and the localized spin of ~5.9 μB on each manganese atom has antiferromagnetic exchange interaction with its nearest neighboring manganese atoms[22,24], as shown in Fig. 1a. Between the layers, each manganese atom has its spin aligned in the same direction with the two manganese atoms directly below and above it. The ratio between in-plane and out-of-plane exchange coupling for the nearest neighbor Mn atoms is ~405:1[24], making the magnons in MnPS₃ highly two-dimensional.

Figure 1b shows the atomic force micrograph of a typical MnPS₃ magnon valve device. The magnon valve constitutes the channel material MnPS₃, an injector, a gate electrode, and a detector. The thickness of the particular MnPS₃ channel is measured to be 12 nm and all electrodes are made of 250 nm wide and 9 nm thick platinum wires. A low-frequency AC current $I_{in}$ is applied to the injector that locally heats up the MnPS₃ crystal and thermally generates diffusive magnons. An in-plane magnetic field is applied that has a component perpendicular to the platinum detector electrode (defined as the x direction, see Fig. 1c) in order to tilt the out-of-plane spins towards such direction. In this configuration, the magnons would carry magnetic moments that can generate a second harmonic nonlocal voltage $V_{2\omega}$ at the platinum detector electrode via the inverse spin Hall effect[10,25]. An additional platinum electrode is fabricated between the

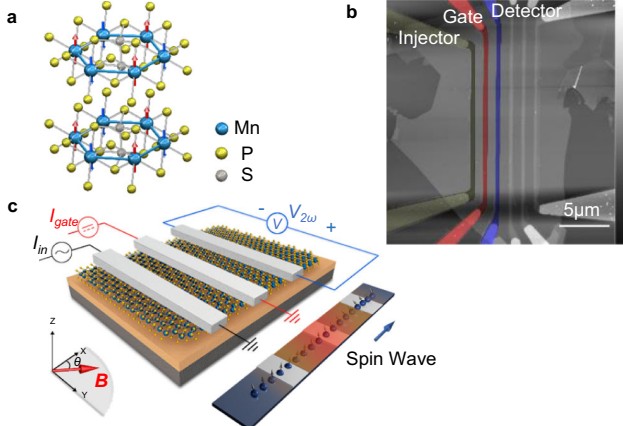

**Fig. 1 Basics of a MnPS₃ magnon valve. a** Atomic model of the crystal and spin structures of antiferromagnetic insulator MnPS₃. **b** Atomic force micrograph of an MnPS₃ magnon valve device. The injector, gate, and detector electrodes are marked by dark green, red, and blue. **c** Artistic schematics of the thermal magnon generation, manipulation, and detection. The upper left section shows the device structure with external circuits and direction of the external in-plane magnetic field; the lower right section shows propagation and modification of spin waves by the gate. Specifically, $I_{in}$: AC injection current; $I_{gate}$: DC gate current; $V_{2\omega}$: the second harmonic thermal magnon inverse spin Hall signal; $\theta$: the angle of the in-plane magnetic field with respect to the x direction.

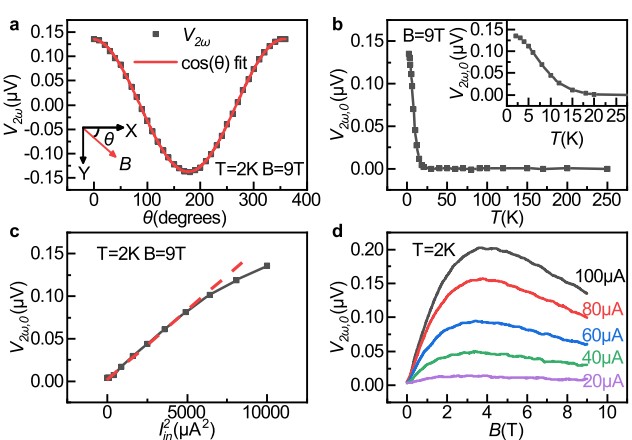

**Fig. 2 Second harmonic magnon signal $V_{2\omega}$ at zero gate current. a** $V_{2\omega}$ as a function of the angle $\theta$ between the external magnetic field **B** and the x direction. Here angle $\theta$ is determined the same as shown in Fig. 1c. **b** Temperature dependence of $V_{2\omega}$ at $\theta = 0$ ($V_{2\omega,0}$). (Inset: zoom-in view of $V_{2\omega,0}$ low-temperature behavior). **c** $V_{2\omega,0}$ versus the square of the injection current $I_{in}^2$. **d** $V_{2\omega,0}$ versus **B** at different $I_{in}$.

injector and detector to act as a gate. We use a DC current $I_{gate}$ applied through the gate electrode to control the signal at the detector. The schematics of the thermal magnon generation, manipulation, and detection are shown in Fig. 1c.

Figure 2a shows the magnetic field angle-dependent $V_{2\omega}$ of a MnPS₃ magnon valve at $T = 2$ K. Here the gate electrode is electrically floating and no current is applied to it. The root mean square value of the injector current $I_{in}$ is 100 μA with a frequency of 18.07 Hz. The external magnetic field of 9 T is applied in-plane with an angle $\theta$ ($\theta = 0$ when $H$ is along the x direction, as shown in Fig. 1c). The red solid line in Fig. 2a is a fit to a cosine function. It can be seen that the $V_{2\omega}(\theta)$ data fits to the cosine function well with a $2\pi$ periodicity and a maximum value at $\theta = 0$, consistent with previous studies on thermal magnons[10,26]. We simplify

$V_{2\omega}(\theta = 0)$ as $V_{2\omega,0}$ in following description. Figure 2b shows the temperature dependence of the $V_{2\omega,0}$. It can be seen that $V_{2\omega,0}$ does not appear until the sample is below 20 K, while the Néel temperature of MnPS$_3$ is around 80 K[21], consistent with previous study[26]. A recent study on thermal magnons in layered ferromagnet CrBr$_3$ also shows similar behavior[27]. The inset in Fig. 2b shows a close up of the rising and saturation behavior of $V_{2\omega,0}$ at low temperature.

Figure 2c shows the dependence of $V_{2\omega,0}$ as a function of $I_{in}$ at the injection electrode, which has a quadratic dependence $V_{2\omega,0} \propto I_{in}^2$ for $I_{in} < 80$ μA, as expected for signal from thermally generated magnons. The deviation from the quadratic relationship at $I_{in} > 80$ μA could be attributed to the increase of the sample temperature in the channel at the detector electrode. Figure 2d shows $V_{2\omega,0}$ on **B** with different $I_{in}$. $V_{2\omega,0}$ initially increases linearly with **B**. This is consistent with the fact that the canting of the spins along the $x$ direction is proportional to **B**, when **B** is small compares to the effective magnetic field of 106 T for the exchange interactions between the nearest neighboring Mn atoms[28]. At higher **B**, $V_{2\omega,0}$ declines slightly, and the peak of $V_{2\omega,0}$ appears to monotonically increase to higher **B** at higher $I_{in}$, which could be owing to a suppression of the magnon diffusion length by external magnetic field[29]. It is worth noting that magnons injected by exchange interactions are absent (i.e., there is zero first harmonic nonlocal signal $V_{1\omega}(\theta)$ with a $\pi$ periodicity to the angle of the in-plane magnetic field) in our MnPS$_3$ devices (see Supplementary Fig. S6), which is consistent with previous studies[26].

We now turn to the gating effect on the thermal magnon signal $V_{2\omega,0}$ with **B** along the $x$ direction. Figure 3a shows a typical dependence of the $V_{2\omega,0}$ on the DC gate current $I_{gate}$. First, it can be seen that $V_{2\omega,0}$ is an even function of $I_{gate}$, e.g., the effects of $+I_{gate}$ and $-I_{gate}$ are identical. Second, there exists a shut-off gate current $I_{gate} = I_0$ that could completely suppress $V_{2\omega,0}$. Remarkably, for $I_{gate} = I_0$, $V_{2\omega,0}$ become negative, and for a sufficiently large gate current $I_{gate} = I_0'$, $V_{2\omega,0}$ tends to zero from the negative side. This means that the thermal magnon signal at the detector can be completely turned off by the gate, at two different values of $I_{gate}$, which is $I_0$ and $I_0'$. The existence of two zero points at the $V_{2\omega,0}(I_{gate})$ curve could be particularly useful for magnon logic operations. Among the two zero points, $I_0$ is more favorable since it is smaller and energetically more efficient.

Figure 3b shows the magnetic field angle-dependent $V_{2\omega}(\theta)$ for different $I_{gate}$. The $V_{2\omega}(\theta)$ curves change sign between the two

$I_{gate}$ ranges: $0 < I_{gate} < I_0$ and $I_0 < I_{gate} < I_0'$. Furthermore, the $V_{2\omega}$ signal at $I_{gate} = I_0$ and $I_{gate} = I_0'$ are suppressed completely for every angle $\theta$. Such experimental observation proves that: (1) the sign reversal of $V_{2\omega,0}$ observed in Fig. 3a can be completely attributed to the thermal magnon signal of the device; (2) the zero points of $V_{2\omega,0}(I_{gate})$ observed in Fig. 3a are indeed zero points of the magnitude of the second harmonic magnon signal. Additional experiment and finite element analysis have also been carried out to rule out the possibility of a local spin Seebeck effect or an anomalous Nernst effect in our devices (details in Supplementary Information S9–S11). If one put $V_{2\omega}$ in a complex coordinate system (e.g., in the $x+iy$ plane, where i is the imaginary unit), $V_{2\omega}$ initially located at the positive side of the $y$ axis with zero gate; as the gate current increases, $V_{2\omega}$ continuously move to the origin along the $y$ axis, reaching the origin at $I_{gate} = I_0$, and continue to move towards the negative side of the $y$ axis with a larger gate; with large enough gate current, e.g., $I_{gate} = I_0'$, $V_{2\omega}$ asymptotically moves back to the origin of the complex 2D plane.

By repeatedly applying $I_{gate} = 0$ and $I_{gate} = 150$ μA, $V_{2\omega,0}$ toggles between 196 nV (On state) and 0 nV (Off state), as shown in Fig. 3c. This demonstrates that the magnon valve can be electrically switched on and off without changing **B**. Such a fully electrical switching of the magnon signal lays the foundation of complex magnonic applications, such as logic gates that mimic those built from the charge-based transistors. In fact, Fig. 3c already illustrates the operation of a diffusive magnon-based NOT gate, which shows finite output ($V_{2\omega,0} = 196$nV) at zero input ($I_{gate} = 0$) and zero output ($V_{2\omega,0} = 0$, here "0" means below the noise floor of our measurement system, which is < 1 nV) at finite input ($I_{gate} = 150$ μA).

In order to understand the gate-dependent behavior of $V_{2\omega,0}$, it is necessary to look into the general form of the inverse spin Hall voltage $V_{ISHE}$ at the detector electrode, which contains $V_{2\omega,0}$ as it is excited by an AC current of frequency $\omega$. $V_{ISHE}$ is proportional to the non-equilibrium magnon accumulation $n_{mag}$ at the detector-MnPS$_3$ interface[30,31]:

$$V_{ISHE}\left(I_{in}(t), I_{gate}\right) \propto g_{mix} n_{mag}(T). \tag{1}$$

Here, $g_{mix}$ is the spin mixing conductance at the detector-MnPS$_3$ interface which can be considered as a constant for our experimental conditions[32]. The non-equilibrium magnon accumulation $n_{mag}$ is caused by the thermally driven magnon spin current $\mathbf{J}_m$ along the $x$ direction due to the Joule heating from the injector and the gate, which in turn is proportional to the lateral temperature gradient in the MnPS$_3$ plane via the spin Seebeck effect[33–36]:

$$n_{mag}(T) \propto \left|\mathbf{J}_m(T)\right| \propto \left|\mathbf{S}(T) \cdot \nabla T\right|. \tag{2}$$

where the $2 \times 2$ spin Seebeck coefficient tensor **S** is defined within the MnPS$_3$ plane and is given below in Eq. (3). Since the temperature increase near the detector is proportional to the Joule heating from current applied in the injector and the gate, the magnon temperature $T = 2K + \beta\left(\alpha I_{in}^2 + I_{gate}^2\right)$, where 2 K is the base temperature for the sample, and $\beta\left(\alpha I_{in}^2 + I_{gate}^2\right)$ accounts for the temperature increase due to the Joule heating from the injector and the gate. Here $\alpha < 1$ is a dimensionless geometrical factor to count for a larger distance of the injector than the gate to the detector, and $\beta$ is a constant involving the resistance of Pt bar and the specific heat of MnPS$_3$. The lateral temperature gradient $\nabla T$ has a similar proportionality as $\nabla T \propto \beta(\alpha I_{in}^2 + I_{gate}^2)\hat{\mathbf{x}}$.

The spin Seebeck coefficient tensor **S** in MnPS$_3$ under in-plane magnetic field can be derived based on a semi-classical

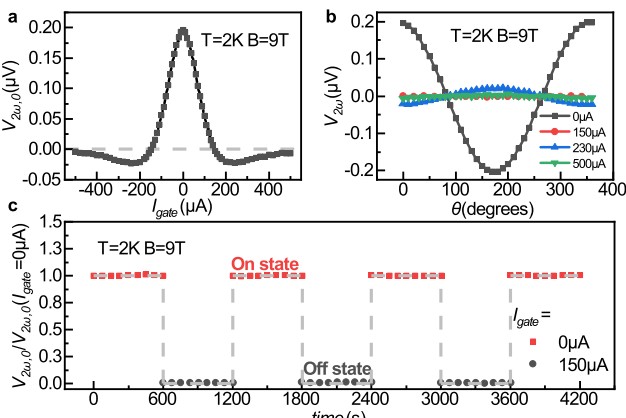

**Fig. 3 Operation of an MnPS$_3$ magnon valve. a** $V_{2\omega,0}$ versus DC gate current $I_{gate}$ at $B = 9$ T and temperature of 2 K. **b** $V_{2\omega}$ as a function of angle $\theta$ of the external magnetic field at four different $I_{gate}$ values. **c** Operation of the MnPS$_3$ magnon valve with $I_{gate}$ toggles repeatedly between 0 μA (On state) and 150 μA (Off state).

Boltzmann transport theory of 2D magnons (details at Supplementary information S3 and S4)[33]:

$$\mathbf{S}(T) = \frac{\hbar^2 \sin\psi}{k_B T^2} \sum_{j=1,2} \int_{BZ} \frac{dk_x dk_y}{(2\pi)^2} \mathbf{v}_j(\mathbf{k}) \mathbf{v}_j(\mathbf{k}) \cosh\xi_j \frac{e^{\hbar\omega_j(\mathbf{k})/k_B T} \omega_j(\mathbf{k})}{\eta_{j,k}\left(e^{\hbar\omega_j(\mathbf{k})/k_B T} - 1\right)^2}$$

(3)

where $\eta_{j,k} = 1/\tau_{j,k}$, $\hbar\omega_j(\mathbf{k})$ and $\mathbf{v}_j(\mathbf{k})$ are the magnon scattering rate, dispersion relation, and group velocity, respectively, for the $j$th magnon branch at magnon momentum $\mathbf{k}$; $\psi$ is the canting angle of the spins from its easy axis at finite in-plane magnetic field, and $\sin\psi\cosh\xi_j$ is the $x$-component spin polarization of magnon density of the $j$th magnon branch (see Supplementary information S4).

Therefore, $V_{ISHE}$ can be regarded as a function of $\beta\left(\alpha I_{in}^2 + I_{gate}^2\right)$, and the temporal dependence of $V_{ISHE}$ comes purely from the time variation of $I_{in}^2(t)$. Obviously, the temperature gradient $\nabla T$ increases monotonically with $\beta\left(\alpha I_{in}^2 + I_{gate}^2\right)$. The behavior of the spin Seebeck coefficient $\mathbf{S}(T)$ as shown in Eq. (3) is not so simple, but qualitatively $\mathbf{S}(T)$ should decrease as function of $\beta\left(\alpha I_{in}^2 + I_{gate}^2\right)$, because the elevated temperature would strongly reduce the magnon mean-free length. Consequently, the thermally driven magnon spin current $\mathbf{J_m}$ will first increase, and then decrease with a general input current. In our real-time lock-in measurement, the first part will give a positive signal since more magnons are accumulating below the detector electrode with a non-zero input from the injection electrode. While in the second decreasing part, less magnons are accumulating below the detector electrode with applying the injection current, which equals to magnons flowing away from the detector electrode, resulting in a negative signal according to ISHE. The simulated functional dependence of $V_{ISHE}$, $\mathbf{S}(T)$ and $\nabla T$ on a general input current can be found in Supplementary Fig. S4a; the temporal dependence and the frequency distribution of $V_{ISHE}$ under an AC excitation are shown in Supplementary Fig. S4b and S4c, respectively.

With the above discussion, we arrive at the following equation from which the performance of the magnon valve devices at different $I_{in}$ and $I_{gate}$ can be simulated out of three global parameters:

$$V_{2\omega,0} = C * \left[\beta\left(\alpha I_{in}^2 + I_{gate}^2\right) * S\left(T = 2K + \beta\left(\alpha I_{in}^2 + I_{gate}^2\right)\right)\right]_{2\omega}$$

(4)

where $C$, $\alpha$, $\beta$ are the three global parameters, and $[\ldots]_{2\omega}$ means taking the second harmonic component.

Figure 4 shows the simulated $V_{2\omega,0}$ vs. $I_{gate}$ curves with $I_{in} = 20$, 40, 60, 80, 100 μA and $\mathbf{B} = 4T$ together with the experimental $V_{2\omega,0}$ data under the same operation conditions. With just three global parameters in Eq. (4) ($C$, $\alpha$, and $\beta$), the simulation reproduces very well the experimental observation, including the naturally inferred symmetric $V_{2\omega,0}$ for $\pm I_{gate}$, the magnitude of $V_{2\omega,0}$ vs. $I_{gate}$ under different $I_{in}$, as well as the values of the zero point of $V_{2\omega,0}$, e.g., $I_0$ and $I_0'$ vs different $I_{in}$. The values of the global parameters $C = 1.05 \times 10^{-26} V \cdot s/\hbar$, $\alpha = 0.25$ and $\beta = 1.69 \times 10^{-4} K/\mu A^2$ are consistent with our experimental configuration, as detailed in Supplementary information S5. Furthermore, the simulation gives vanishingly small first harmonic response $V_{1\omega,0}$, which agrees with the physics of thermal magnon excitation and it is indeed what we observed experimentally (see Supplementary Fig. S6). The above agreement between the simulation and the experimental data suggests that our model captures the physical trends behind the switching behavior of the MnPS$_3$ magnon valves. The zero points at the thermal magnonic signal is guaranteed from our general argument, which comes from the highly tunable magnon spin current of the van der Waals antiferromagnetic

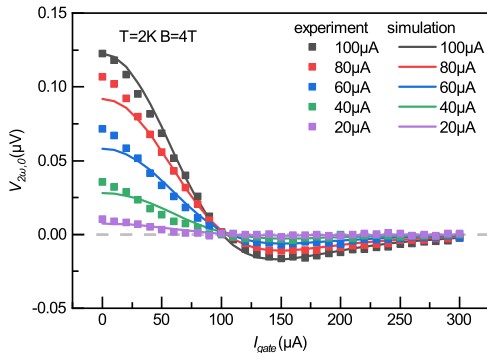

**Fig. 4 Gate tuning of a MnPS$_3$ magnon valve with different $I_{in}$: simulation and experiments.** Simulation of gate-dependent $V_{2\omega,0}$ at 2 K and 4 T with five selected injection current $I_{in}$. Only three global parameters are needed to produce all the simulation curves which match well with the experimental data.

insulators via electrical means. It is interesting to note that the first zero crossing point $I_0$ is a weak function of the injection current $I_{in}$, as can be seen in Fig. 4, which can be understood as the consequence of the steep initial slop and sharp peak of the inverse spin Hall voltage $V_{ISHE}$ with an input current (as can be seen in Supplementary Fig. S4a). Indeed, the simulated zero crossing point $I_0$ also reveals its weak but finite dependence on the injection current $I_{in}$ (Supplementary Fig. S8). It is also worth noting that, albeit good overall agreement is found between the experimental data and theoretical simulation with only three global parameters, appreciable difference between the experiment and simulation is found for small $I_{gate}$ (i.e., $I_{gate} < 50$ μA), which hints the existence of additional factors and warrants further study.

In conclusion, electrically controlled MnPS$_3$-based magnon valves have been realized, in which the second harmonic magnon signal can be turned off by DC current through a metal gate. Such a zero crossing demonstrates a complete blocking of the transmission of the second harmonic magnon signal, enabled by the nonlinear gate dependence of the non-equilibrium magnon density of the van der Waals antiferromagnetic crystal. We expect that other van der Waals antiferromagnetic insulators with weak interlayer interactions would have very similar properties. Such strong and reversible electrical control of the magnon signal demonstrates the potential of van der Waals insulators with magnetic orders and paves the way to application of magnonics in future digital circuits.

## Methods

**Device fabrication and sample characterization**. MnPS$_3$ flakes are mechanically exfoliated from bulk crystals and deposited on 300 nm SiO$_2$/Si substrates. Thickness of the MnPS$_3$ flakes used in our study ranges from 10 nm to 30 nm. The injector, gate and detector electrodes in the magnon valves are fabricated with standard electron-beam lithography, platinum deposition and lift-off processes. Platinum is deposited in a magnetron sputtering system, and the width of the wires is ~250 nm with a thickness of 9 nm. Afterwards, 5 nm of titanium and 80 nm of gold are patterned to contact the platinum wires. Twelve devices (device 1–12) were made and studied. Data shown in the main text were obtained from device 1 (Fig. 1–3) and device 2 (Fig. 4), and the results for other devices are presented in the Supplementary Information.

**Nonlocal magnon transport measurement**. The magnon transport measurement in MnPS$_3$ is done in a physical properties measurement system (PPMS) with low-frequency lock-in amplifier technique. The injection AC current (18.07 Hz) in the range from 0 μA to 100 μA is provided by lock-in amplifier (NF LI5640) or a signal generator (Tektronix AFG 3000) with a 10KΩ resistor. Lock-in amplifiers (Stanford Research SR830) are used to probe the nonlocal voltages. Low noise voltage preamplifiers (NF LI75A) are also used. A voltage source (Keithley 2400) with a 100KΩ resistor is used to provide the DC current (0 μA~500 μA) to modulate the nonlocal signal. The temperature of the measurement in PPMS ranges from 2 K to 300 K, the applied magnetic field is parallel to our sample plane and the maximum field is 9 T.

## Data availability
Data for figures that support the current study are available at https://doi.org/10.7910/DVN/GLESFN. Source data are provided with this paper.

## Code availability
Numerical simulations were performed with Mathematica. Source codes in Mathematica file format are available from the corresponding authors upon reasonable request. Source data are provided with this paper.

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

## Acknowledgements
This project has been supported by the National Basic Research Program of China 2019YFA0308402, 2019YFA0308401, 2018YFA0305604, 2015CB921102, the National Natural Science Foundation of China 11934001, 11774010, 11921005, Beijing Municipal Natural Science Foundation JQ20002, the Strategic Priority Research Program of Chinese Academy of Sciences XDB28000000, National Research Foundation Singapore program NRF-CRP21-2018-0007, NRF-CRP22-2019-0007 and Singapore Ministry of Education via AcRF Tier 3 Program 'Geometrical Quantum Materials' MOE2018-T3-1-002.

## Author contributions
J.-H.C. conceived and coordinated the experiment; G.C. and S.Q. fabricated the devices and performed most of the measurement; S.Y. aided in transport measurement; Y.Z. aided in device fabrication and AFM measurement; M.L., J.L., J.X., X.C., and R.S. provided theoretical analysis; J.L. and R.S. performed the analysis of spin Seebeck coefficient; K.C. performed fitting to various possible mechanisms; S.Q., G.C., and D.C. performed simulations according to J.L and R.S.'s analysis; P.Y. and Z.L. have grown high-quality MnPS$_3$ bulk crystals; J.-H.C. wrote the manuscript and all authors commented and modified the manuscript.

## Competing interests
The authors declare no competing interests.
