## [Peer Review File · Nature Communications]

Reviewers' Comments:

Reviewer #1:

Remarks to the Author:

The manuscript reports new type of magnon valves based on few-layer antiferromagnet MnPS₃, which can be turned on and off through purely electric method. More specifically, the authors first excited magnons with Joule heating of injector electrode and detected non-equilibrium magnons at the detector through inverse spin Hall effect. While this part has been demonstrated recently in reference 26, the authors then designed a gate electrode between injector/detector and successfully manipulate the thermal magnon signal through the Joule heating effect of gate electrode, clean and repeatable on/off states have been demonstrated. The authors further developed a simple theoretical model which fitted well the experimental data.

van der Waals 2D magnets attract intense attention recently and have been used in different types of spintronic devices, this work represents the first one in the form of magnon valve. The design is novel and the data is beautiful, this work thus opens a new route to utilize 2D magnets in spintronics and would generate tremendous interest in the research community. So I recommend the publication in Nature Communications. However, there are several points need to be addressed before publication:

1. The key achievement of this work is tuning nonlocal voltage from positive to negative value so that zero-voltage state is guaranteed, but the underlying physical mechanism is not clearly presented in the main text. From the formula 4 of the main text, it is not straightforward why the nonlocal voltage will become negative as all numbers are positive. Besides that, why the crossing point (from positive to negative) is the same for different injection current (Fig. 4)? The crossing of nonlocal voltage would relate to competition between temperature gradient produced by gating electrode and inject electrode, more insightful points could be obtained from the analysis of the fitting process and should be discussed.
2. Why the nonlocal voltage vanishes above 20K? The Neel temperature of bulk MnPS₃ is around 80K, it should not be so much different for the investigated flakes, given the flakes are not too thin and MnPS₃ is van der Waals magnet with strong anisotropy. It is also experimentally demonstrated that Neel temperature of thin flakes does not change much in recent studies (such as the one in ref. 21).
3. As several devices were presented in main text and supplementary information, it should be clearly marked in figures or captions which device the presented data belongs to.

Reviewer #2:

Remarks to the Author:

This work shows the modulation of the second-harmonic nonlocal signal in thin MnPS₃ flakes through a gate electrode. The authors show the magnetic field and in-plane angle dependence of their signals which are consistent with thermally generated magnons and detection via the spin Seebeck effect. The nonlocal signals are shown to be modulated by a gate electrode between the Pt-heater and Pt detector, which shows a modulation on the nonlocal second-harmonic signal, symmetric around zero current. The modulation on the signals is very large, crossing zero for currents around 150 μ A.

This work will be of interest to the 2D materials community working on magnon transport. However, as I detail below, I do not believe it has a high enough impact nor are the conclusions backed up by their data to ensure publication in Nature Communications. In my opinion a revised version of this work would be better suited to a more topical journal such as Phys. Rev. Applied or Phys. Rev. B.

- 1) One of my main concerns is that there is no data supporting magnon transport and all their results could be explained via a local spin Seebeck (or anomalous Nernst) effect. The gate modulation on their signals is thermal in nature and it could be explained by a change in the local temperature gradients below the detector contact. In order to claim magnon transport, the authors should at least provide a careful heater-detector distance dependence and back-up their results through a proper (finite-element) modeling of the temperature gradients. Moreover, it would be relevant to provide the modulation of the signal for different values of injection (heating) current.

At the moment it looks coincidental that the crossover point in gate current is nearly equal to the injection current, however I believe this is due to the (local) temperature profile created by the two electrodes. Finally, the authors should provide a measurement where the gate electrode is located on the opposite side of the injector, with the configuration heater-detector-gate. This will provide further insight on the temperature profile created by the injector and gate electrodes.

2) A similar sign reversal of the spin Seebeck signal with the injector heating current has been reported before for the 2D ferromagnet CrBr₃ [Phys. Rev. B 101, 205407 (2020)]. The shape of the signal modulation has a striking resemblance to the ones reported here, and can be explained by the simple heat profile under the detector contact and the inclusion of the anomalous Nernst effect. The authors should take this into account and provide a discussion along these lines, including/excluding the possible effects and making a proper comparison with previous experiments.

3) The authors never mention the presence or absence of electrically-injected magnons, i.e. a first-harmonic signal. It should be at least mentioned that the authors did not see a first-harmonic signal above their noise floor, and give a value for that, so an upper bound for a possible signal is given.

4) The angular dependence of the signals shows that the spin Seebeck signal is proportional to the net magnetization induced by the field due to the canting of the magnetic moments. However, line 124 of page 5 gives the impression that this is due to a change of the antiferromagnetic magnon modes in MnPS₃, which should appear with the field applied along the Néel vector. The authors should clarify this point.

5) What is the sign of the signal shown here compared to previous results in MnPS₃ [ref. 26], YIG [ref. 10] and CrBr₃ [Phys. Rev. B 101, 205407 (2020)]? The positive and negative voltage probes should be clearly indicated in Fig. 1.

6) In order to extract a possible contribution from the anomalous Nernst effect the authors should also present an out-of-plane magnetic field angular dependence. This would also help clarifying the heat gradient directions in their system.

Reviewer #3:

Remarks to the Author:

In this manuscript, the authors have claimed the realization of a magnon valve based on van der Waals anti-ferromagnetic insulator few layers MnPS₃. They show the tunability of the second harmonic non-local Voltage by applying a DC current through a metal "gate" located between the injection and detection electrodes. The behavior of the non-local signal is then simulated using the spin Seebeck coefficient based on a semi-classical Boltzmann transport theory with 3 free parameters.

Although the characteristic features of thermal magnon transport seems reasonably convincing I have several comments to be addressed by the authors:

1) It is quite common that magnetic vdW crystals are sensitive to ambient conditions and most of the time reactive to oxygen or humidity exposure. While some vdW materials such as most of the TMDs can resist to even rather high temperature (300degC), it has been shown that MnPS₃ would degrade. Do the authors have any way of confirming the integrity and quality of the topmost layers of their crystals? Would encapsulation be an option to avoid any damages?

2) The non-local signal is discussed as arising from the temperature gradient generated by the injection and the detection electrodes and modulated by the "gate" electrode. I am not sure to understand fully the microscopic mechanism of the "gating" electrode. Can the authors give a more detailed description? Could it be that the suppression of the non-local signal is simply due to an overheating of the junction with the gate electrode as observed at higher temperatures (above 20K)?

3) I am not very convinced by the simulations obtained by fitting 3 free parameters. Although the discussion on the obtained values can have some credit in my point of view, I believe that some other external parameters could be used to refine or remove some parameters. One example could be to look at the "gate" dependence of the ratio $V_{2w,in} / V_{2w,Gate}$ (as described in Supplementary) for various temperature and find a relation between alpha and beta.

4) Other minor typos and coments:

- Line 57 vdw magnets have weak...
- Line 152 I believe the Off states should be limited by the resolution of the measurement setup and not exactly 0 nV.
- Fig 3b on the y-axis it should be V_{2w} and not $V_{2w,0}$

Reviewer #1 (Remarks to the Author):

The manuscript reports new type of magnon valves based on few-layer antiferromagnet MnPS₃, which can be turned on and off through purely electric method. More specifically, the authors first excited magnons with Joule heating of injector electrode and detected non-equilibrium magnons at the detector through inverse spin Hall effect. While this part has been demonstrated recently in reference 26, the authors then designed a gate electrode between injector/detector and successfully manipulate the thermal magnon signal through the Joule heating effect of gate electrode, clean and repeatable on/off states have been demonstrated. The authors further developed a simple theoretical model which fitted well the experimental data.

van der Waals 2D magnets attract intense attention recently and have been used in different types of spintronics devices, this work represents the first one in the form of magnon valve. The design is novel and the data is beautiful, this work thus opens a new route to utilize 2D magnets in spintronics and would generate tremendous interest in the research community. So I recommend the publication in Nature Communications. However, there are several points need to be addressed before publication:

We are grateful that the reviewer pointed out the major findings and the significance of our work. We have made substantial modifications to our manuscript according to the comments of our reviewers and we believe we have successfully addressed all the questions and comments from the reviewers. Our point-by-point reply is listed below.

1. The key achievement of this work is tuning nonlocal voltage from positive to negative value so that zero-voltage state is guaranteed, but the underlying physical mechanism is not clearly presented in the main text. From the formula 4 of the main text, it is not straightforward why the nonlocal voltage will become negative as all numbers are positive. Besides that, why the crossing point (from positive to negative) is the same for different injection current (Fig. 4)? The crossing of nonlocal voltage would relate to competition between temperature gradient produced by gating electrode and inject electrode, more insightful points could be obtained from the analysis of the fitting process and should be discussed.

We thank the reviewer for pointing out the part of our manuscript that needs to be clarified. $V_{2\omega}$ is an AC non-local inverse spin Hall signal with frequency 2ω , where ω is the frequency of the driving AC injector current. Equation (4) in the main text means that the second harmonic component $V_{2\omega}$ consists of two time-varying components, i.e. S and ∇T . Namely, the Seebeck coefficient S depends on the average temperature (a mean temperature between the injector and detector), which we consider to depend on the time due to time-dependent heating by the AC injector current. The negative sign appears when these two time-varying components are in the opposite phase. More generally, $V_{2\omega}$ is complex-valued. If we put the data in a complex coordinate system (e.g., in the $x+iy$ plane), the AC signal $V_{2\omega}$ initially

located at the positive side of the y axis with zero gate; as the gate current increases, $V_{2\omega}$ continuously move to the origin along the y axis, reaching the origin at $I_{\text{gate}} = I_0$, and continue to towards the negative side of the y axis with larger gate; with large enough gate current, e.g. $I_{\text{gate}} = I_0'$, $V_{2\omega}$ asymptotically moves back to the origin of the complex 2D plane. According to the reviewer's comment, we have added new discussion in page 7, line 158 of the revised main text: "If one put $V_{2\omega}$ in a complex coordinate system (e.g., in the $x+iy$ plane, where i is the imaginary unit), $V_{2\omega}$ initially located at the positive side of the y axis with zero gate; as the gate current increases, $V_{2\omega}$ continuously move to the origin along the y axis, reaching the origin at $I_{\text{gate}} = I_0$, and continue to towards the negative side of the y axis with larger gate; with large enough gate current, e.g. $I_{\text{gate}} = I_0'$, $V_{2\omega}$ asymptotically moves back to the origin of the complex 2D plane." The modification is marked in yellow in the revised main text.

Next, the reviewer is correct that the crossing point is a weak function of the injection current, thus it looks like that they all cross at one point. If we look closely, the crossing point is indeed dependent on the injection current. We have plotted the simulated crossing point I_0 vs. injection current I_{in} bellow as Figure R1 (and as revised Supplementary Figure S8), which clearly shows the finite dependence of I_0 as a function of I_{in} . Here, the fact that $I_0 \sim 100 \mu\text{A}$ while $I_{\text{in}} \sim 100 \mu\text{A}$ should be a coincidence which is related to the sample thickness, electrode parameters, etc.

Figure R1(Figure S8). The simulated crossing point I_0 (the first I_{gate} value for $V_{2\omega} = 0$) vs. injection current I_{in} . The simulation is based on parameters obtained from fitting to experimental data shown in Figure 4 in the main text.

According to the reviewer's comment, we have modified our manuscript at page 11, line 245 as: "It is interesting to note that the first zero crossing point I_0 is a weak function of the injection current I_{in} , as can be seen in Figure 4, which can be understood as the consequence of the steep initial slope and sharp peak of the inverse spin Hall voltage V_{ISHE} with an input current (as can be seen in Supplementary Figure S4a). Indeed, the simulated zero crossing point I_0 also reveals its weak but finite dependence on the injection current I_{in} (Supplementary Figure S8)." We have also added a new Supplementary Figure S8 with description: "The simulated crossing point I_0 (the first I_{gate} value for $V_{2\omega} = 0$) vs. injection current I_{in} . The simulation is based on parameters obtained from fitting to experimental data shown in Figure 4 in the main text."

2. Why the nonlocal voltage vanishes above 20K? The Neel temperature of bulk MnPS₃ is around 80K, it should not be so much different for the investigated flakes, given the flakes are not too thin and MnPS₃ is van der Waals magnet with strong anisotropy. It is also experimentally demonstrated that Neel temperature of thin flakes does not change much in recent studies (such as the one in ref. 21).

This is a very insightful question. We have plenty of experimental proof that the non-local voltage appears below ~20K, but does not show up around the Néel temperature. This phenomenon is also independently confirmed by other groups, such as Ref. 26 in our manuscript. As a matter of fact, we experimentally found that other van der Waals magnets have similar behaviors from a number of other works we are currently writing up. In the literature, recent experiment on CrBr₃ (PRB 101, 205407 (2020)) also shows such unusual behavior. It would be interesting to carry out future works in clarifying the exact cause of such behavior. We have modified our manuscript to reflect the reviewer's comments at page 5, line 115 in the revised manuscript: "It can be seen that $V_{2\omega,0}$ does not appear until the sample is below 20 K, while the Néel temperature of MnPS₃ is around 80K[21], consistent with previous study²⁶. A recent study on thermal magnons in layered ferromagnet CrBr₃ also shows similar behavior²⁷."

3. As several devices were presented in main text and supplementary information, it should be clearly marked in figures or captions which device the presented data belongs to.

We thank the reviewer for the thoughtful advice. We have modified our manuscript to clearly mark the different devices appeared in our main text and in the supplementary information.

Reviewer #2 (Remarks to the Author):

This work shows the modulation of the second-harmonic nonlocal signal in thin MnPS₃ flakes through a gate electrode. The authors show the magnetic field and in-plane angle dependence of their signals which are consistent with thermally generated magnons and detection via the spin Seebeck effect. The nonlocal signals are shown to be modulated by a gate electrode between the Pt-heater and Pt detector, which shows a modulation on the nonlocal second-harmonic signal, symmetric around zero current. The modulation on the signals is very large, crossing zero for currents around 150 μ A.

This work will be of interest to the 2D materials community working on magnon transport. However, as I detail below, I do not believe it has a high enough impact nor are the conclusions backed up by their data to ensure publication in Nature Communications. In my opinion a revised version of this work would be better suited to a more topical journal such as Phys. Rev. Applied or Phys. Rev. B.

We thank the reviewer for acknowledging that 100% modulation of non-local second harmonic signal has been achieved in our magnon valve devices made from van der Waals anti-ferromagnetic insulator MnPS₃. We also thank the reviewer for the helpful comments which help us to greatly improve our manuscript. We have carried out extensive experiment and analysis which we believe have successfully addressed all the concerns raised by the reviewer.

We respectfully argue that our work is highly important as it is the first diffusive magnon valve that can be turned off completely and electrically. It paves the way for magnon binary logic which is readily interfaced with charge-based logics. What's more, our work is a discovery of a general characteristic of the highly tunable two-dimensional magnon system, which we already found that it could be applied to a number of van der Waals anti-ferromagnets.

1) One of my main concerns is that there is no data supporting magnon transport and all their results could be explained via a local spin Seebeck (or anomalous Nernst) effect. The gate modulation on their signals is thermal in nature and it could be explained by a change in the local temperature gradients below the detector contact. In order to claim magnon transport, the authors should at least provide a careful heater-detector distance dependence and back-up their results through a proper (finite-element) modeling of the temperature gradients. Moreover, it would be relevant to provide the modulation of the signal for different values of injection (heating) current. At the moment it looks coincidental that the crossover point in gate current is nearly equal to the injection current, however I believe this is due to the (local) temperature profile created by the two electrodes. Finally, the authors should provide a measurement where the gate electrode is located on the opposite side of the injector, with the configuration heater-detector-gate. This will provide further insight on the temperature profile created by the injector and gate electrodes.

The reviewer raised the question that the data may be explained by either a local spin Seebeck effect or anomalous Nernst effect without assuming any magnon transport from the injector to the detector. Namely, in the “local spin Seebeck” scenario, the heat transport from the injector electrode to the detector electrode is carried by phonon instead of by magnon. The heat around the detector electrodes causes the longitudinal spin Seebeck effect, which injects magnon or spin into the detector electrode. In the “anomalous Nernst” scenario, the Pt detector electrode is magnetized along the in-plane field direction, and the temperature gradient across the interface induces an inverse spin Hall voltage in the detector electrode. In these two scenarios of the reviewer’s, the magnon transport from the detector electrode to the injector electrode is not assumed.

Firstly, we would like to thank the reviewer for bringing up the need to clarify between magnon transport and phonon transport. We also thank the reviewer for pointing out the effect of variation for the location of injector electrode, gate electrode and detector electrode. Inspired by the reviewer’s comments, we have done a number of experiment as well as corresponding finite element analysis for the temperature gradient in the system, which unambiguously proved that the signal we measured are NEITHER the local spin Seebeck effect NOR the anomalous Nernst effect.

MnPS₃ is highly electrically insulating. Thus there are only two possibilities that an in-plane magnetic field dependent second harmonic non-local signal can be detected in our device configuration: magnon transport and phonon transport (the later should be aided by local spin Seebeck/anomalous Nernst effect to be detected in our device configuration). Both effects would have second harmonic non-local signal that have a $\cos\theta$ dependence on the angle θ between the magnetic field and the direction perpendicular to the detector Pt electrode.

Nonetheless, compares to CrBr₃ where inside each layer the magnetic coupling is ferromagnetic, and YIG where there are net magnetization due to the ferrimagnetic nature of the material, MnPS₃ is has Ising type anti-ferromagnetic order within each Mn atomic layer (see Figure R2 below). Thus, it is much less likely that the Pt/MnPS₃ interface would be magnetized via proximity effect. This material properties of MnPS₃ makes us believe that the anomalous Nernst effect to be unlikely; while in the following we have three strong evidence that excluded anomalous Nernst effect or local spin Hall effect to affect our results.

Figure R2. The schematic of the interface of Pt/MnPS₃, Pt/CrBr₃ and Pt/YIG.

Here we provide three additional experimental evidences that excluded the possibility of local spin Seebeck effect as well as anomalous Nernst effect:

- a) **To clarify whether the heat is carried by phonon or by magnon**, we have fabricated a non-local device with a number of electrodes on MnPS₃. As depicted in Figure R3a below, we name four of the electrodes as Detector 1, Injector, oxidized Cu strip and Detector 2, respectively. All electrodes are made from Pt except for the oxidized Cu strip. The oxidized Cu strip is made from 10nm thick copper without any protection capping layer and then is exposed to ambient condition for oxidation. We have made sure that the Cu strip was conductive right after the deposition and not conductive after oxidation. The intention of the oxidation is to reduce the thermal conductivity of the copper strip to 4 W/m*K [A. Kusiak et al. Eur. Phys. J. Appl. Phys 35, 17-27(2006)], so that it is much lower than a Pt electrode in terms of thermal conductivity (72W/m*K for Pt). The oxidized copper strip merely acts as surface absorbates which only affect the top surface of MnPS₃ and would not act as a strong heat sink. In another word, the oxidized Cu strip should perturb magnon transport much more than phonon transport in MnPS₃, since the in-plane to out-of-plane ratio of magnetic coupling strength is 405:1 [JPCM 10, 6417-6428 (1998)] while the in-plane to out-of-plane ratio of thermal conductivity is only 6:1[ACS Nano 14, 2424-2435 (2020)]. The strong in-plane versus out-of-plane anisotropy in the magnetic exchange suggests that the Pt detectable magnon transport goes through only a few top layers of the sample, while the phonon transport generally goes through the whole layers of the sample. Being only the surface absorbates, the oxidized copper strip is expected to perturb the magnon transport dramatically, while the phonon transport remains robust against such perturbation.

An AC signal is applied through the Injector electrode, and the signal is measured simultaneously from Detector 1 and Detector 2. We found strong signal from Detector 1 (right next to the Injector) and no signal from Detector 2 (the oxidized copper strip is between Detector 2 and the Injector). Since the temperature gradient between Injector and Detector 2 should be finite as the case for Detector 1, which is confirmed by finite element analysis (see Figure R3c). The absence of the non-local inverse spin Hall signal from Detector 2 proves that phonon transport is not the cause of the non-local signal from the Detector electrode. We have added this data in the revised Supplementary Figure S9, with a short description.

Figure R3 (Figure S9). Non-local magnon signal detection with different MnPS₃ device geometries. (a) Schematic of non-local measurement on a MnPS₃ device with oxidized Cu strip. (b) Temperature dependence of $V_{2\omega}$ at $\theta = 0$ for the detector located at the left or right side of the oxidized Cu strip. (c) Finite element analysis of the temperature distribution in MnPS₃ device with oxidized Cu strip.

- b) **To quantify the effect of the anomalous Nernst effect in our experimental system**, we have measured the non-local second harmonic signal with an applied magnetic field of up to 14 T rotated in the x - z plane (see inset in Figure R4a below). Since there is finite temperature gradient along the x axis from our device configuration as shown from the finite element analysis, the temperature gradient along x also induces the Hall voltage along the y axis in the presence of the magnetization along the z axis (ANEx). According to PRB 101, 205407, ANEx and ANEz are of similar magnitude, where ANEz refers to the Hall voltage along y induced by the temperature gradient along z in the presence of the magnetization along the x axis (ANEz).

The angle of the magnetic field with respect to the z axis is marked as φ . An injection current of 100 μA is applied to the injector of our MnPS₃ device. We can see from Figure R4a that the data fits well to a $\sin\varphi$ function, in which the signal is zero when the magnetic field is along the z axis (perpendicular to the sample plane). From Figure R4a one can also see that only the magnetic field component along the x axis could produce non-zero non-local second harmonic signal. Figure R4b shows minimal magnetic-field dependence of the non-local signal with the magnetic field along the z axis (i.e., $\varphi=0$). This data

proves unambiguously the absence of anomalous Nernst effect with magnetic field perpendicular to the MnPS₃/Pt interface (ANEx), because the finite element calculation shows a finite temperature gradient along x . The absence of ANEx points to the absence of ANEz.

In fact, MnPS₃ is a layer antiferromagnet where the spin within one Mn atomic layer is aligned antiferromagnetically with a coupling constant J that amount to about 106T of magnetic field, which far exceeds the magnetic field applied in the experiment. It is natural that the MnPS₃/Pt interface remains non-magnetized. We have added this data in Supplementary Figure S10, with a short description.

Figure R4 (Figure S10). The absence of the anomalous Nernst effect in MnPS₃ magnon valve. (a) The non-local second harmonic signal as a function of angle ϕ between the external magnetic field ($B=9T$) and the z direction, angle ϕ is determined the same as shown in the inset.(b) The absence of magnetic field dependence of $V_{2\omega}$ at $\phi = 0$.

- c) **Distance dependent measurement on our MnPS₃ device** (see Figure R5 below). The decay length of the second harmonic thermal magnon signal is measured to be $\sim 3300\text{nm}$, which is consistent with previous report in the literature (e.g. $\sim 2800\text{ nm}$ for 16-nm MnPS₃, 1100 nm for 8-nm MnPS₃ as reported in PRX 9, 011026 (2019)). As shown in Fig. R5, this decay length is much longer than a decay length of the temperature gradient from the finite element calculation that represents how far the phonon carries the heat in space. **Thus, the longer decay length in the heater-detector distance dependence suggests that the signal cannot be explained by the phonon transport.** We have added the data in revised Supplementary Information S11.

Figure R5 (Figure S11). The heater-detector distance-dependent signal and temperature in MnPS₃ device. Left axis: The distance-dependent non-local second harmonic signal $V_{2\omega,0}$ and relaxation length fitting [Ref. 10] at 2K and 9T; Right axis: The finite element simulation of distance-dependent temperature of MnPS₃ under the same device configuration. The decay length of the second harmonic thermal magnon signal is measured to be about 3300nm.

Second, about the relative location of gate electrode. We would also like to stress that our theoretical model is based on the magnon band structure of MnPS₃ which provides the Seebeck coefficient for magnon transport in our device. Based on this model, the non-local voltage is a direct consequence of the spin current injected into the detector electrode given finite temperature gradient in the device provided by the injector electrode and the gate electrode. There is no restriction on the relative location of the gate and the injector in order to make our magnon valve work. The advantage of our model is that the details of the temperature gradient is absorbed into the three global parameters C , α and β , which are fixed for any particular device geometry operating at a certain base temperature and magnetic field. We found that in the injector-detector-gate configuration, the general behavior is similar (see Figure R6b). Finite element analysis shows that the variation of the temperature gradient is also similar for the gate located at the right or left side of the detector electrode (see Figure R6c&d), while the temperature of the MnPS₃ below the detector electrode is slightly higher for the case of the injector-detector-gate configuration, which may be the cause of the slight difference observed experimentally. We have added this data in Supplementary Figure S12, with a short explanation.

Figure R6 (Figure S12). Operation of a MnPS₃ magnon valve with different device geometries. (a) Schematics of non-local measurement on a MnPS₃ device with different gates. (b) $V_{2\omega,0}$ versus DC gate current I_{gate} at $B = 9T$ and temperature of 2K with different geometries. (c)(d) Finite element analysis of the temperature distribution in MnPS₃ device for the gate located at the left (c) or right (d) side of the detector electrode.

The parameters used in the finite element analysis are listed below:

Pt	Conductivity	8.9E6[S/m]	COMSOL Material database
	thermal conductivity	71.6[W/(m*K)]	COMSOL Material database
MnPS ₃	in-plane thermal conductivity	6.3[W/(m*K)]	ACS Nano,14, 2424–2435(2020)
	through-plane thermal conductivity	1.1[W/(m*K)]	ACS Nano,14, 2424–2435(2020)

SiO ₂	thermal conductivity	1.38[W/(m*K)]	CRC Handbook of Chemistry and Physics (92nd ed.).p12.213
Si	thermal conductivity	130[W/(m*K)]	COMSOL Material database
MnPS ₃ /SiO ₂	through-plane thermal resistance	5E-7[K*m ² /W] [#]	Computational Materials Science, 142, 1–6 (2018)
Pt/MnPS ₃	through-plane thermal resistance	1.4E-7[K*m ² /W] [§]	PHYSICAL REVIEW B 101, 205407 (2020)

Table R1. The parameters used in the finite element analysis. [#]There is no data found for MnPS₃/SiO₂ in the literature, we used value from through-plane thermal resistance between MoS₂/SiO₂ instead. [§]There is no data found for Pt/MnPS₃, we used estimated value for CrBr₃/Pt in the literature.

2) A similar sign reversal of the spin Seebeck signal with the injector heating current has been reported before for the 2D ferromagnet CrBr₃ [Phys. Rev. B 101, 205407 (2020)]. The shape of the signal modulation has a striking resemblance to the ones reported here, and can be explained by the simple heat profile under the detector contact and the inclusion of the anomalous Nernst effect. The authors should take this into account and provide a discussion along these lines, including/excluding the possible effects and making a proper comparison with previous experiments.

We would like to thank the reviewer for pointing out a potentially confusing point. It is a very good opportunity for us to clarify the three significant differences between our work and the work in PRB 101, 205407 (2020):

First of all, the channel materials are very different. MnPS₃ is a layered anti-ferromagnet with Ising-type anti-ferromagnetic coupling in the sample plane, while CrBr₃ is a 2D ferromagnet. This means that there is zero magnetic moment in each layer of MnPS₃, while each layer of CrBr₃ is magnetic. Such difference reflects strongly in the magnon spectra of the two materials [PRX 8, 011010 (2018), PRB 103, 024424 (2021)], and also reflects strongly in the existence of the magnetization at the Pt-CrBr₃ interface and the absence of which at the Pt-MnPS₃ interface, resulting in the observation of a large anomalous Nernst signal in CrBr₃ (PRB 101, 205407) and the absence of which in our work (see discussions in Reviewer #2, comment #1).

Second, the $R_{2\omega}$ vs. I_{in} curves are very different. We have plotted the $R_{2\omega}$ vs. I_{in} in Figure R7 (revised Supplementary Figure S13) for a couple of MnPS₃ devices below. In order to compare with the work on CrBr₃, the gate electrodes in our devices are floating during the measurement. It can be seen that the shape of the $R_{2\omega}$ vs. I_{in} of MnPS₃ device is very different from that of the CrBr₃ device shown in PRB 101, 205407. Interestingly, the shape of the $R_{2\omega}$ vs. I_{in} for CrBr₃ device has some

resemblance with the $V_{2\omega}$ vs. I_{in} for our MnPS₃ devices, which would be a good topic for future works.

Third, the devices are very different. Our work realized the first diffusive magnon valves in which a gate current controls whether the injected signal can be detected or not, which readily enables digital logic operation; PRB 101, 205407 (2020) describe a non-local response curve for the input signal without any external gate control.

According to the reviewer's comment, we have added a new section (S7) in the revised Supplementary Information to discuss the differences between our work and the work in PRB 101, 205407 (2020). The added discussion is colored in blue in the revised Supplementary Information.

Figure R7 (Figure S13). The $R_{2\omega}$ vs. I_{in} for different MnPS₃ devices with zero gate current.

3) The authors never mention the presence or absence of electrically-injected magnons, i.e. a first-harmonic signal. It should be at least mentioned that the authors did not see a first-harmonic signal above their noise floor, and give a value for that, so an upper bound for a possible signal is given.

We thank the reviewer for pointing out the need to mention the absence of electrically-injected magnons (first harmonic signal) in the beginning of the main text. As a matter of fact, we have written in page 10, line 237: “Furthermore, the simulation gives vanishingly small first harmonic response $V_{1\omega,0}$, which agrees with the physics of thermal magnon excitation and it is indeed what we observed experimentally (see Supplementary Fig. S6).” We have plotted the absence of the first-harmonic signal together with the finite second harmonic signal in the original Supplementary Figure S6. To improve the visibility of our discussion on electrically-injected magnons, we have added the following discussion at page 6, line 131 of the revised manuscript: “It is worth noting that magnons injected by exchange interactions are absent (i.e., there is zero first harmonic non-local signal $V_{1\omega}(\theta)$ with a π periodicity to the angle of the in-plane magnetic field) in our MnPS₃ devices (see Supplementary Figure S6), which is consistent with previous studies [PRX 9, 011026 (2019)].” The added discussion is highlighted in yellow in the revised manuscript.

4) The angular dependence of the signals shows that the spin Seebeck signal is proportional to the net magnetization induced by the field due to the canting of the magnetic moments. However, line 124 of page 5 gives the impression that this is due to a change of the antiferromagnetic magnon modes in MnPS₃, which should appear with the field applied along the Néel vector. The authors should clarify this point.

We thank the reviewer for bring up this point that might cause confusion. Indeed, when the applied magnetic field B is along the Néel vector, the energy of the magnon modes will be changing linearly with B ; when the applied B field is perpendicular to the Néel vector, the energy of the magnon modes will be changing quadratic with B . However, such linearity or quadratic relation between magnon energy and the magnetic field produces higher order effects in $V_{2\omega}$ as compares to the magnetization from the canting of the spins in the Mn layers. Specifically, from Eq. (3) at page 8, line 185 of the main text (page 9, line 200 in the revised main text), we can see that the linearity comes from the $\sin\psi$ term located at the beginning of Eq. (3). Here ψ is the canting angle of the spins from its easy axis at finite in-plane magnetic field. For small magnetic field, the canting angle is proportional to the in-plane magnetic field. We have modified our manuscript at page 5, line 126 to clarify this point: “This is consistent with the fact that the canting of the spins along the x direction is proportional to B , when B is small compares to the effective magnetic field of 106T for the exchange interactions between the nearest neighboring Mn atoms²⁸.”

Note that in the previous version of the manuscript, we used θ to denote the canting angle. For clarity, we have now changed the Greek letter for canting angle from θ to ψ

at page 9, line 200, line 203 and line 204 in the revised main text, due to the fact that we have already defined θ as the angle of the in-plane magnetic field with respect to the Pt electrode. The Greek letter representing the canting angle is also changed from θ to ψ in the revised Supplementary Information. All the revision and addition are highlighted in yellow in the main text and colored in blue in the Supplementary Information.

5) What is the sign of the signal shown here compared to previous results in MnPS₃ [ref. 26], YIG [ref. 10] and CrBr₃ [Phys. Rev. B 101, 205407 (2020)]? The positive and negative voltage probes should be clearly indicated in Fig. 1.

We thank the reviewer for bringing up the definition of the sign of the signal. We have been very careful about getting the right sign as well, which can be summarized below in Figure R8. Our definition of positive and negative voltage probes is completely the same as those in previous works on MnPS₃ [Ref. 26], on CrBr₃ [PRB 101, 205407 (2020)] and on YIG [Ref. 10] (for non-local signal with distance larger than ~250 nm in Ref. 10). According to the reviewer's comment, we have modified Fig. 1 by indicating the positive and negative voltage probes. Here we would like to mention that PRB 101, 205407 (2020) measured a "negative" spin Seebeck signal because the "positive" direction of the magnetic field they used is the opposite of ours (our definition of a positive magnetic field is shown in Figure R8), while the definition of positive and negative voltage probes are the same for both papers.

Figure R8. Schematic of non-local measurement on a typical MnPS₃ device via the low-frequency lock-in technique.

6) In order to extract a possible contribution from the anomalous Nernst effect the authors should also present an out-of-plane magnetic field angular dependence. This would also help clarifying the heat gradient directions in their system.

As discussed in the response to question #1 of reviewer #2, we have measured the second harmonic signal with vertical magnetic field of up to 14 T and an injection current of 100 μ A in our MnPS₃ device, and could not detect any non-local second harmonic signal, drastically different from the case of in-plane magnetic field

perpendicular to the Pt electrode. In conjunction with results in PRB 101, 205407 (2020), this proves the absence of anomalous Nernst effect in our MnPS₃ device. We have added this data in Supplementary Figure S10, with a short explanation.

Reviewer #3 (Remarks to the Author):

In this manuscript, the authors have claimed the realization of a magnon valve based on van der Waals anti-ferromagnetic insulator few layers MnPS₃. They show the tunability of the second harmonic non-local Voltage by applying a DC current through a metal “gate” located between the injection and detection electrodes. The behavior of the non-local signal is then simulated using the spin Seebeck coefficient based on a semi-classical Boltzmann transport theory with 3 free parameters.

Although the characteristic features of thermal magnon transport seem reasonably convincing I have several comments to be addressed by the authors:

We are glad that the reviewer found our result convincing and we appreciate the reviewer’s comments that help us substantially improve our manuscript.

1) It is quite common that magnetic vdW crystals are sensitive to ambient conditions and most of the time reactive to oxygen or humidity exposure. While some vdW materials such as most of the TMDs can resist to even rather high temperature (300degC), it has been shown that MnPS₃ would degrade. Do the authors have any way of confirming the integrity and quality of the topmost layers of their crystals? Would encapsulation be an option to avoid any damages?

Indeed a lot of vdW crystals are quite sensitive to ambient conditions. Many of the vdW magnetic materials, including but not limited to CrI₃, CrBr₃, Fe₃GeTe₂, etc., can degrade quickly when exposed to air. Similarly, the temperature stability is also an issue for a lot of vdW crystals. In the beginning of our experiment, we have carefully checked the air stability and thermal stability of MnPS₃. We found that few-layer MnPS₃ is quite air stable. We have taken the optical micrograph of few-layer MnPS₃ on SiO₂ substrate right after exfoliation and 8 months in air after exfoliation, which has no visible changes (see figure R9a below, also in the revised Supplementary Figure S14a). We found that few-layer MnPS₃ is also thermally stable up to 350 degC in air, which is well above the highest temperature (150 degC) during our device fabrication process. We have put few-layer MnPS₃ in a hot plate and have taken optical micrographs before heating and after heating to 350 degC in air (see figure R9b below, also in the revised Supplementary Figure S14b), where no visible changes are found. Thus, the stability of MnPS₃ is another advantage of the material as compares to many less stable layered magnetics materials.

Figure R9 (Figure S14). Stability test of few-layer MnPS₃. (a) the optical micrograph of few-layer MnPS₃ on SiO₂ substrate right after exfoliation and 8 months after exfoliation. (b) optical micrographs of few-layer MnPS₃ on SiO₂ substrate before heating and after heating to 150, 250 and 350°C for 10 minutes in air. (c) The device performance of our MnPS₃ magnon device right after fabrication and after 8 months.

That being said, optical micrographs can only tell us whether serious degradations happened. So we still paid special attention in handling the samples, and make sure that the sample does not unnecessarily expose to ambient conditions. We have

measured few-layer MnPS₃ devices right after fabrication and 8 months after fabrication, and did not see substantial degradation in the signal quality. We have put the data in the revised Supplementary Figure S14c and also in Figure R9c for your convenience. Encapsulation with BN is an effective method to improve the device stability for the highly reactive vdW crystals. Thus, using BN encapsulation should also provide MnPS₃ devices with more protection, but will at the same time increase the difficulty of a good contact between MnPS₃ and the Pt electrode. A very good two-dimensional contact is needed for the detector electrode to receive spin angular momentum injection from the non-equilibrium MnPS₃ magnons. Since the devices work well without encapsulation, we employ the precaution of avoiding prolonged exposure of the device to air and of storing the devices in Argon environment while they are not being measured or processed. This procedure works well for us in the experiment with MnPS₃.

2) The non-local signal is discussed as arising from the temperature gradient generated by the injection and the detection electrodes and modulated by the “gate” electrode. I am not sure to understand fully the microscopic mechanism of the “gating” electrode. Can the authors give a more detailed description? Could it be that the suppression of the non-local signal is simply due to an overheating of the junction with the gate electrode as observed at higher temperatures (above 20K)?

We thank the reviewer for bringing up this important point. In the beginning of our experiment, we have indeed been trying to attribute the tuning effect of the gate current to simply the thermal effect that heat up the whole crystal to temperature higher than 20K. However, the fact that: 1) there are two zero points for gate current I_0 and I_0' , and 2) between I_0 and I_0' , the second harmonic voltage changes sign as compares to gate current smaller than I_0 , cannot be explained by such picture. The microscopic mechanism of the “gating” electrode lies in the line shape of V_{ISHE} shown in Fig. S4 (a). We have added more discussion in page 9, line 212 to clarify the microscopic mechanism of the non-local signal: “Consequently, the thermally driven magnon spin current J_m will first increase, and then decrease with a general input current. In our real-time lock-in measurement, the first part will give a positive signal since more magnons are accumulating below the detector electrode with a non-zero input from the injection electrode. While in the second decreasing part, less magnons are accumulating below the detector electrode with applying the injection current, which equals to magnons flowing away from the detector electrode, resulting in a negative signal according to ISHE.” Also, from our finite element simulation, our sample temperature has never reached temperatures higher than 20K, which is consistent with our theoretical model discussed above.

3) I am not very convinced by the simulations obtained by fitting 3 free parameters. Although the discussion on the obtained values can have some credit in my point of view, I believe that some other external parameters could be used to refine or remove some parameters. One example could be to look at the “gate” dependence of the ratio

$V_{2\omega,in}/V_{2\omega,Gate}$ (as described in Supplementary) for various temperature and find a relation between alpha and beta.

The reviewer brings up an important point worth careful discussion: could we further reduce the parameters in the simulation. The discussion of these parameters can be found in Supplementary Section S5, we shall discuss them in more details here:

1) The overall parameter C in equation (4) mainly includes factors that relates a spin current injected into the detector Pt electrode to the voltage generated in the electrode; it also includes the dimensionless prefactor of the magnon relaxation time $1/\eta_{j,k}$ after summing contributions from all the magnon bands j and the momentum k in the first Brillouin Zone, which brings in various effects including crystal quality, temperature and magnetic field. Thus this parameter is difficult to be removed yet very easily determined via fitting to an experimental curve.

2) The parameter β characterizes the effectiveness for the injection and gating current to be converted to temperature gradient in the device. In principle it could be obtained from parameters such as the resistivity of the Pt strips, the thermal conductivity of the Pt strips and the Au electrodes that are connected to the Pt strips (the 200nm Pt strips are connected to Cr/Au electrodes which extended into bonding pads for the devices), the thermal conductivity of the MnPS₃ crystal, as well as thermal resistivity of all the material interfaces. Considering the complexity of all these external parameters and our goal to derive a predictive effective model, the parameter β is best to be determined via fitting to an experimental curve.

3) The parameter α is the ratio of the effective strength of the injector electrode over the gate electrode in terms of their influence to the detector-MnPS₃ interface. This ratio is also device dependent, mainly depending on the device structure, the channel length and thickness, as well as the material of the channel and the electrodes. If equation (4) is a linear function of the temperature gradient, parameter α could be the easiest to be obtained from additional tests, such as the one we did in Supplementary Information S5 (e.g., measure $V_{2\omega}$ for $I_{in} = 100\mu A$ and $I_{gate} = 0\mu A$ and $I_{in} = 0\mu A$ and $I_{gate,ac} = 100\mu A$). However, since S is an integral function that contains the temperature, and it is highly non-linear as shown in Supplementary Figure S4a, there is no simple relation between $V'_{2\omega,in}/V'_{2\omega,gate}$ and α . Thus, it is still the most straightforward to use three parameters in the simulation. We have extended the discussion in the revised Supplementary Information S5 to clarify this issue, and the revised discussion is colored in blue.

4) Other minor typos and comments:

- Line 57 vdw magnets have weak...
- Fig 3b on the y-axis it should be $V_{2\omega}$ and not $V_{2\omega,0}$

We thank the reviewer for pointing out the typos. We have corrected the typos in the revised manuscript. The corrections are highlighted in yellow in the revised main text.

- Line 152 I believe the Off states should be limited by the resolution of the measurement setup and not exactly 0 nV.

We agree with the reviewer that there is always a limit as to how accurately we could measure “0”; we would also like to stress that since the signal goes from positive to negative, the zero point (for the physical quantity itself, not the noise-limited measured values we could get) is guaranteed. To reflect the comments from the reviewer, we have modified page 7, line 169 in the revised manuscript to be: “In fact, Fig. 3c already illustrates the operation of a diffusive magnon based NOT gate, which shows finite output ($V_{2\omega} = 196$ nV) at zero input ($I_{\text{gate}} = 0$) and zero output ($V_{2\omega} = 0$), here “0” means below the noise floor of our measurement system, which is < 1 nV) at finite input ($I_{\text{gate}} = 150\mu\text{A}$).”

Reviewers' Comments:

Reviewer #1:

Remarks to the Author:

The authors have satisfactorily addressed all issues I raised in my last report, and I think they also reasonably clarified the questions of other reviewers through extended experiments and simulations. So I recommend the publication of current manuscript in Nature Communications.

Reviewer #2:

Remarks to the Author:

In their response, updated manuscript, and supplementary information the authors bring up new data and analysis which address the questions and concerns raised by me and the other two reviewers. My main previous concern was that the magnon transport and modulation shown is solely of local origin (i.e. local spin Seebeck effect without magnon transport). With the new experiments, simulations and arguments, I am convinced that this is likely not the case. Indeed, the authors do a very careful job addressing my comments, which I appreciate. I find the new version of the manuscript much more thorough and attractive to researchers in magnonics and magnetic 2D materials. Therefore, I believe this manuscript can now be published in Nature Communications.

Nevertheless, before publication I would still like to ask the authors to address the following points:

1) The discussion brought up in the response letter on my main point (local versus non-local magnon detection) is overlooked in the new version of the supplementary material. Even though the figures are added, there is no text accompanying it. I understand that the response letters are also published, but I would strongly suggest the authors to add the discussion accompanying figures R3 to R6 and table R1 to the supplementary information.

2) I would also suggest the authors to include the fitting results of Fig. R5 (S11) with their respective errors.

3) In the response and new version of the SI, the authors report that their V_{2w} versus I_{in} curves look similar to the R_{2w} vs I_{in} for CrBr₃ [ref 27], but the curves reported for R_{2w} vs I_{in} (Fig. R7 / S13) do not show a sign reversal of R_{2w} as in Ref 27. Here I am assuming that $R_{2w} = V_{2w}/(I_{in})^2$ as it is conventionally used in literature. Perhaps the authors mean R_{2w} vs I_{in} of Ref. 27 compared to their V_{2w} vs I_{gate} ?

Reviewer #3:

Remarks to the Author:

The authors have answered most of my remarks satisfactorily. However, I am still not convinced about the bold statement of the authors "strongly" suggesting that their simulation "captures the main physics behind the switching behavior of the MnPS₃ magnon valves" based on a fitting with 3 free parameters which seems in rather poor agreement at low I_{gate} ($I_{gate} < 50 \mu A$). The authors have brought good argument concerning why the different parameters are indeed complex to be obtained from an experimental point of view but the validity of a model should not depend on such consideration. In my opinion, I think that the authors should lower their claims concerning the simulation.

Reviewer #1 (Remarks to the Author):

The authors have satisfactorily addressed all issues I raised in my last report, and I think they also reasonably clarified the questions of other reviewers through extended experiments and simulations. So I recommend the publication of current manuscript in Nature Communications.

We are glad that the reviewer considered our revision satisfactory and recommend the publication of our current manuscript in Nature Communications. We highly appreciate the helpful comments from the reviewer that helped us greatly improve our manuscript.

Reviewer #2 (Remarks to the Author):

In their response, updated manuscript, and supplementary information the authors bring up new data and analysis which address the questions and concerns raised by me and the other two reviewers. My main previous concern was that the magnon transport and modulation shown is solely of local origin (i.e. local spin Seebeck effect without magnon transport). With the new experiments, simulations and arguments, I am convinced that this is likely not the case. Indeed, the authors do a very careful job addressing my comments, which I appreciate. I find the new version of the manuscript much more thorough and attractive to researchers in magnonics and magnetic 2D materials. Therefore, I believe this manuscript can now be published in Nature Communications.

We are grateful for the questions raised by the reviewer, which prompted us to substantially improve our manuscript through additional experiment, simulation and analysis. Our point-by-point reply to the additional comments of the reviewer is listed below. All revisions in this round of review process are marked in blue in the revised manuscript and supplementary information.

Nevertheless, before publication I would still like to ask the authors to address the following points:

- 1) The discussion brought up in the response letter on my main point (local versus non-local magnon detection) is overlooked in the new version of the supplementary material. Even though the figures are added, there is no text accompanying it. I understand that the response letters are also published, but I would strongly suggest the authors to add the discussion accompanying figures R3 to R6 and table R1 to the supplementary information.

We agree with the reviewer that the supplementary information should contain more text explanation to facilitate the understanding of our work by the readers, as people

might not regularly look into the response letters. We have added the following explanations:

1) In Supplementary Information Section 9:

“Section S9 to S12 contain additional experimental evidences that excluded the possibility of local spin Seebeck effect as well as anomalous Nernst effect in the MnPS_3 magnon valves.

First of all, to clarify whether the heat is carried by phonon or by magnon, we have fabricated a non-local device with a number of electrodes on MnPS_3 . As depicted in Fig. S9 below, we name four of the electrodes as Detector 1, Injector, oxidized Cu strip and Detector 2, respectively. All electrodes are made from Pt except for the oxidized Cu strip. The oxidized Cu strip is made from 10nm thick copper without any protection capping layer and then is exposed to ambient condition for oxidation. We have made sure that the Cu strip was conductive right after the deposition and not conductive after oxidation. The intention of the oxidation is to reduce the thermal conductivity of the copper strip to 4 $\text{W/m}\cdot\text{K}$ [10], so that it is much lower than a Pt electrode in terms of thermal conductivity (72 $\text{W/m}\cdot\text{K}$ for Pt). The oxidized copper strip merely acts as surface absorbates which only affect the top surface of MnPS_3 and would not act as a strong heat sink. In another word, the oxidized Cu strip should perturb magnon transport much more than phonon transport in MnPS_3 , since the in-plane to out-of-plane ratio of magnetic coupling strength is 405:1 [1] while the in-plane to out-of-plane ratio of thermal conductivity is only 6:1 [11]. The strong in-plane versus out-of-plane anisotropy in the magnetic exchange suggests that the Pt detectable magnon transport goes through only a few top layers of the sample, while the phonon transport generally goes through the whole layers of the sample. Being only the surface absorbates, the oxidized copper strip is expected to perturb the magnon transport dramatically, while the phonon transport remains robust against such perturbation.

An AC signal is applied through the Injector electrode, and the signal is measured simultaneously from Detector 1 and Detector 2. We found strong signal from Detector 1 (right next to the Injector) and no signal from Detector 2 (the oxidized copper strip is between Detector 2 and the Injector). Since the temperature gradient between Injector and Detector 2 should be finite as the case for Detector 1, which is confirmed by finite element analysis (see Fig. S9c). The absence of the non-local inverse spin Hall signal from Detector 2 proves that phonon transport is not the cause of the non-local signal from the Detector electrode.”

2) In Supplementary Information Section 10:

“To quantify the effect of the anomalous Nernst effect in our experimental system, we have measured the non-local second harmonic signal with an applied magnetic field of up to 14 T rotated in the x - z plane (see inset in Fig. S10 below). Since there is finite temperature gradient along the x axis from our device configuration as shown from the finite element analysis, the temperature gradient along x also induces the Hall voltage along the y axis in the presence of the magnetization

along the z axis (ANEx). It's considered that ANEx and ANEz are of similar magnitude¹², where ANEz refers to the Hall voltage along y induced by the temperature gradient along z in the presence of the magnetization along the x axis. The angle of the magnetic field with respect to the z axis is marked as φ . An injection current of 100 μA is applied to the injector of our MnPS₃ device. We can see from Fig. S10a that the data fits well to a $\sin\varphi$ function, in which the signal is zero when the magnetic field is along the z axis (perpendicular to the sample plane). From Fig. S10a one can also see that only the magnetic field component along the x axis could produce non-zero non-local second harmonic signal. Figure S10b shows minimal magnetic-field dependence of the non-local signal with the magnetic field along the z axis (i.e., $\varphi=0$). This data proves unambiguously the absence of anomalous Nernst effect with magnetic field perpendicular to the MnPS₃/Pt interface (ANEx), because the finite element calculation shows a finite temperature gradient along x . The absence of ANEx points to the absence of ANEz¹².

In fact, MnPS₃ is a layer antiferromagnet where the spin within one Mn atomic layer is aligned antiferromagnetically with a coupling constant J that amount to about 106T of magnetic field, which far exceeds the magnetic field applied in the experiment. It is natural that the MnPS₃/Pt interface remains non-magnetized. We have added this data in Supplementary Figure S10, with a short description.”

3) In Supplementary Information Section 11:

“We have measured the distance dependence of $V_{2\omega,0}$ (black dots in Fig. S11) and the experimental data is fitted to $V_{2\omega} = \frac{C_0}{\lambda} * \frac{\exp(d/\lambda)}{1-\exp(2d/\lambda)}$ [13], where C_0 is a factor characterizing the magnitude of the second harmonic signal, λ is the decay length of the diffusive magnons. The fitting gives $C_0 = -2.4\pm 0.2 \mu\text{V}$ and $\lambda = 3300\pm 200$ nm, which is consistent with previous report in the literature (e.g. $\lambda\sim 2800$ nm for 16-nm MnPS₃, 1100 nm for 8-nm MnPS₃ [14]). As shown in Fig. S11, this decay length (red curve) is much longer than a decay length of the temperature gradient from the finite element calculation (blue curve) that represents how far the phonon carries the heat in space. Thus, the longer decay length in the heater-detector distance dependence suggests that the signal cannot be explained by the phonon transport.”

4) In Supplementary Information Section 12:

“Two different device geometries, namely, the injector-gate-detector configuration as well as the injector-detector -gate configuration is tested. We found that in the injector-detector-gate configuration, the general behavior is similar (see Fig. S12b). Finite element analysis shows that the variation of the temperature gradient is also similar for the gate located at the right or left side of the detector electrode (see Fig. S12c&d), while the temperature of the MnPS₃ below the detector electrode is slightly higher for the case of the injector-detector-gate configuration,

which may be the cause of the slight difference in $V_{2\omega}$ vs. I_{gate} observed experimentally.”

5) Supplementary Table S1 is added into the supplementary information after the text shown in 4) above.

2) I would also suggest the authors to include the fitting results of Fig. R5 (S11) with their respective errors.

We have added discussion in Supplementary Information S11: “We have measured the distance dependence of $V_{2\omega,0}$ (black dots in Fig. S11) and the experimental data is fitted to $V_{2\omega} = \frac{C_0}{\lambda} * \frac{\exp(d/\lambda)}{1-\exp(2d/\lambda)}$ [13], where C_0 is a factor characterizing the magnitude of the second harmonic signal, λ is the decay length of the diffusive magnons. The fitting gives $C_0 = -2.4 \pm 0.2 \mu\text{V}$ and $\lambda = 3300 \pm 200 \text{ nm}$, which is consistent with previous report in the literature (e.g. $\lambda \sim 2800 \text{ nm}$ for 16-nm MnPS₃, 1100 nm for 8-nm MnPS₃ [14]). As shown in Fig. S11, this decay length (red curve) is much longer than a decay length of the temperature gradient from the finite element calculation (blue curve) that represents how far the phonon carries the heat in space. Thus, the longer decay length in the heater-detector distance dependence suggests that the signal cannot be explained by the phonon transport.”

3) In the response and new version of the SI, the authors report that their $V_{2\omega}$ versus I_{in} curves look similar to the $R_{2\omega}$ vs I_{in} for CrBr₃ [ref 27], but the curves reported for $R_{2\omega}$ vs I_{in} (Fig. R7 / S13) do not show a sign reversal of $R_{2\omega}$ as in Ref 27. Here I am assuming that $R_{2\omega} = V_{2\omega}/(I_{\text{in}})^2$ as it is conventionally used in literature. Perhaps the authors mean $R_{2\omega}$ vs I_{in} of Ref. 27 compared to their $V_{2\omega}$ vs I_{gate} ?

We would like to thank the careful reviewing from the reviewer. We realized that it was a typo in our previous response to the reviewer’s question in the first round of review. The sentence: “Interestingly, the shape of the $R_{2\omega}$ vs. I_{in} for CrBr₃ device has some resemblance with the $V_{2\omega}$ vs. I_{in} for our MnPS₃ devices...” should actually be: “Interestingly, the shape of the $R_{2\omega}$ vs. I_{in} for CrBr₃ device has some resemblance with the $V_{2\omega}$ vs. I_{gate} for our MnPS₃ devices...” We apologize for the typo in our previous reply.

Indeed, we use the convention $R_{2\omega} = V_{2\omega}/(I_{\text{in}})^2$ to obtain Fig. S13, thus $R_{2\omega}$ and $V_{2\omega}$ always have the same sign. In order to illustrate this matter better, we added Fig. S14 in the supplementary information showing $V_{2\omega}$ vs. I_{in} for the same set of MnPS₃

devices as shown in Fig. S13. The new Fig. S14 is also reproduced below for the reviewer's convenience. As can be seen in Fig. S14, $V_{2\omega}$ vs. I_{in} for MnPS₃ devices are very different from $V_{2\omega}$ vs. I_{in} for CrBr₃ devices [ref 27].

Fig. S14. The $V_{2\omega}$ vs. I_{in} for the same set of MnPS₃ devices with zero gate current as shown in Fig. S13.

Reviewer #3 (Remarks to the Author):

The authors have answered most of my remarks satisfactorily. However, I am still not convinced about the bold statement of the authors “strongly” suggesting that their simulation “captures the main physics behind the switching behavior of the MnPS₃ magnon valves” based on a fitting with 3 free parameters which seems in rather poor agreement at low I_{gate} ($I_{\text{gate}} < 50 \mu\text{A}$). The authors have brought good argument concerning why the different parameters are indeed complex to be obtained from an experimental point of view but the validity of a model should not depend on such consideration. In my opinion, I think that the authors should lower their claims concerning the simulation.

We agree with the reviewer that some deviation between experiment and simulation is present for low I_{gate} , which means that additional factors are still out there beyond our current model. Part of our model is phenomenological (i.e. Eq. (4)), which certainly warrants further refinement and research in future studies. We are glad that the reviewer brought up this point, and we have made modifications accordingly, in page 11, line 239 of the revised manuscript: “The above agreement between the simulation and the experimental data suggests that our model captures the physical trends behind the switching behavior of the MnPS₃ magnon valves.” We have also added discussion in page 11, line 249 of the revised manuscript: “It is also worth noting that, albeit good overall agreement is found between the experimental data and theoretical simulation with only three global parameters, appreciable difference between the experiment and simulation is found for small I_{gate} (i.e., $I_{\text{gate}} < 50 \mu\text{A}$), which hints the existence of additional factors and warrants further study.” All revisions in this round of review process are marked in yellow in the revised manuscript.

Reviewers' Comments:

Reviewer #2:

Remarks to the Author:

The authors have successfully addressed my comments from my previous report. I can now recommend the publication of this manuscript as is.

Reviewer #1 (Remarks to the Author):

The manuscript reports new type of magnon valves based on few-layer antiferromagnet MnPS₃, which can be turned on and off through purely electric method. More specifically, the authors first excited magnons with Joule heating of injector electrode and detected non-equilibrium magnons at the detector through inverse spin Hall effect. While this part has been demonstrated recently in reference 26, the authors then designed a gate electrode between injector/detector and successfully manipulate the thermal magnon signal through the Joule heating effect of gate electrode, clean and repeatable on/off states have been demonstrated. The authors further developed a simple theoretical model which fitted well the experimental data.

van der Waals 2D magnets attract intense attention recently and have been used in different types of spintronics devices, this work represents the first one in the form of magnon valve. The design is novel and the data is beautiful, this work thus opens a new route to utilize 2D magnets in spintronics and would generate tremendous interest in the research community. So I recommend the publication in Nature Communications. However, there are several points need to be addressed before publication:

We are grateful that the reviewer pointed out the major findings and the significance of our work. We have made substantial modifications to our manuscript according to the comments of our reviewers and we believe we have successfully addressed all the questions and comments from the reviewers. Our point-by-point reply is listed below.

1. The key achievement of this work is tuning nonlocal voltage from positive to negative value so that zero-voltage state is guaranteed, but the underlying physical mechanism is not clearly presented in the main text. From the formula 4 of the main text, it is not straightforward why the nonlocal voltage will become negative as all numbers are positive. Besides that, why the crossing point (from positive to negative) is the same for different injection current (Fig. 4)? The crossing of nonlocal voltage would relate to competition between temperature gradient produced by gating electrode and inject electrode, more insightful points could be obtained from the analysis of the fitting process and should be discussed.

We thank the reviewer for pointing out the part of our manuscript that needs to be clarified. $V_{2\omega}$ is an AC non-local inverse spin Hall signal with frequency 2ω , where ω is the frequency of the driving AC injector current. Equation (4) in the main text means that the second harmonic component $V_{2\omega}$ consists of two time-varying components, i.e. S and ∇T . Namely, the Seebeck coefficient S depends on the average temperature (a mean temperature between the injector and detector), which we consider to depend on the time due to time-dependent heating by the AC injector current. The negative sign appears when these two time-varying components are in the opposite phase. More generally, $V_{2\omega}$ is complex-valued. If we put the data in a complex coordinate system (e.g., in the $x+iy$ plane), the AC signal $V_{2\omega}$ initially

located at the positive side of the y axis with zero gate; as the gate current increases, $V_{2\omega}$ continuously move to the origin along the y axis, reaching the origin at $I_{\text{gate}} = I_0$, and continue to towards the negative side of the y axis with larger gate; with large enough gate current, e.g. $I_{\text{gate}} = I_0'$, $V_{2\omega}$ asymptotically moves back to the origin of the complex 2D plane. According to the reviewer's comment, we have added new discussion in page 7, line 158 of the revised main text: "If one put $V_{2\omega}$ in a complex coordinate system (e.g., in the $x+iy$ plane, where i is the imaginary unit), $V_{2\omega}$ initially located at the positive side of the y axis with zero gate; as the gate current increases, $V_{2\omega}$ continuously move to the origin along the y axis, reaching the origin at $I_{\text{gate}} = I_0$, and continue to towards the negative side of the y axis with larger gate; with large enough gate current, e.g. $I_{\text{gate}} = I_0'$, $V_{2\omega}$ asymptotically moves back to the origin of the complex 2D plane." The modification is marked in yellow in the revised main text.

Next, the reviewer is correct that the crossing point is a weak function of the injection current, thus it looks like that they all cross at one point. If we look closely, the crossing point is indeed dependent on the injection current. We have plotted the simulated crossing point I_0 vs. injection current I_{in} bellow as Figure R1 (and as revised Supplementary Figure S8), which clearly shows the finite dependence of I_0 as a function of I_{in} . Here, the fact that $I_0 \sim 100 \mu\text{A}$ while $I_{\text{in}} \sim 100 \mu\text{A}$ should be a coincidence which is related to the sample thickness, electrode parameters, etc.

Figure R1(Figure S8). The simulated crossing point I_0 (the first I_{gate} value for $V_{2\omega} = 0$) vs. injection current I_{in} . The simulation is based on parameters obtained from fitting to experimental data shown in Figure 4 in the main text.

According to the reviewer's comment, we have modified our manuscript at page 11, line 245 as: "It is interesting to note that the first zero crossing point I_0 is a weak function of the injection current I_{in} , as can be seen in Figure 4, which can be understood as the consequence of the steep initial slope and sharp peak of the inverse spin Hall voltage V_{ISHE} with an input current (as can be seen in Supplementary Figure S4a). Indeed, the simulated zero crossing point I_0 also reveals its weak but finite dependence on the injection current I_{in} (Supplementary Figure S8)." We have also added a new Supplementary Figure S8 with description: "The simulated crossing point I_0 (the first I_{gate} value for $V_{2\omega} = 0$) vs. injection current I_{in} . The simulation is based on parameters obtained from fitting to experimental data shown in Figure 4 in the main text."

2. Why the nonlocal voltage vanishes above 20K? The Neel temperature of bulk MnPS₃ is around 80K, it should not be so much different for the investigated flakes, given the flakes are not too thin and MnPS₃ is van der Waals magnet with strong anisotropy. It is also experimentally demonstrated that Neel temperature of thin flakes does not change much in recent studies (such as the one in ref. 21).

This is a very insightful question. We have plenty of experimental proof that the non-local voltage appears below ~20K, but does not show up around the Néel temperature. This phenomenon is also independently confirmed by other groups, such as Ref. 26 in our manuscript. As a matter of fact, we experimentally found that other van der Waals magnets have similar behaviors from a number of other works we are currently writing up. In the literature, recent experiment on CrBr₃ (PRB 101, 205407 (2020)) also shows such unusual behavior. It would be interesting to carry out future works in clarifying the exact cause of such behavior. We have modified our manuscript to reflect the reviewer's comments at page 5, line 115 in the revised manuscript: "It can be seen that $V_{2\omega,0}$ does not appear until the sample is below 20 K, while the Néel temperature of MnPS₃ is around 80K[21], consistent with previous study²⁶. A recent study on thermal magnons in layered ferromagnet CrBr₃ also shows similar behavior²⁷."

3. As several devices were presented in main text and supplementary information, it should be clearly marked in figures or captions which device the presented data belongs to.

We thank the reviewer for the thoughtful advice. We have modified our manuscript to clearly mark the different devices appeared in our main text and in the supplementary information.

Reviewer #2 (Remarks to the Author):

This work shows the modulation of the second-harmonic nonlocal signal in thin MnPS₃ flakes through a gate electrode. The authors show the magnetic field and in-plane angle dependence of their signals which are consistent with thermally generated magnons and detection via the spin Seebeck effect. The nonlocal signals are shown to be modulated by a gate electrode between the Pt-heater and Pt detector, which shows a modulation on the nonlocal second-harmonic signal, symmetric around zero current. The modulation on the signals is very large, crossing zero for currents around 150 μ A.

This work will be of interest to the 2D materials community working on magnon transport. However, as I detail below, I do not believe it has a high enough impact nor are the conclusions backed up by their data to ensure publication in Nature Communications. In my opinion a revised version of this work would be better suited to a more topical journal such as Phys. Rev. Applied or Phys. Rev. B.

We thank the reviewer for acknowledging that 100% modulation of non-local second harmonic signal has been achieved in our magnon valve devices made from van der Waals anti-ferromagnetic insulator MnPS₃. We also thank the reviewer for the helpful comments which help us to greatly improve our manuscript. We have carried out extensive experiment and analysis which we believe have successfully addressed all the concerns raised by the reviewer.

We respectfully argue that our work is highly important as it is the first diffusive magnon valve that can be turned off completely and electrically. It paves the way for magnon binary logic which is readily interfaced with charge-based logics. What's more, our work is a discovery of a general characteristic of the highly tunable two-dimensional magnon system, which we already found that it could be applied to a number of van der Waals anti-ferromagnets.

1) One of my main concerns is that there is no data supporting magnon transport and all their results could be explained via a local spin Seebeck (or anomalous Nernst) effect. The gate modulation on their signals is thermal in nature and it could be explained by a change in the local temperature gradients below the detector contact. In order to claim magnon transport, the authors should at least provide a careful heater-detector distance dependence and back-up their results through a proper (finite-element) modeling of the temperature gradients. Moreover, it would be relevant to provide the modulation of the signal for different values of injection (heating) current. At the moment it looks coincidental that the crossover point in gate current is nearly equal to the injection current, however I believe this is due to the (local) temperature profile created by the two electrodes. Finally, the authors should provide a measurement where the gate electrode is located on the opposite side of the injector, with the configuration heater-detector-gate. This will provide further insight on the temperature profile created by the injector and gate electrodes.

The reviewer raised the question that the data may be explained by either a local spin Seebeck effect or anomalous Nernst effect without assuming any magnon transport from the injector to the detector. Namely, in the “local spin Seebeck” scenario, the heat transport from the injector electrode to the detector electrode is carried by phonon instead of by magnon. The heat around the detector electrodes causes the longitudinal spin Seebeck effect, which injects magnon or spin into the detector electrode. In the “anomalous Nernst” scenario, the Pt detector electrode is magnetized along the in-plane field direction, and the temperature gradient across the interface induces an inverse spin Hall voltage in the detector electrode. In these two scenarios of the reviewer’s, the magnon transport from the detector electrode to the injector electrode is not assumed.

Firstly, we would like to thank the reviewer for bringing up the need to clarify between magnon transport and phonon transport. We also thank the reviewer for pointing out the effect of variation for the location of injector electrode, gate electrode and detector electrode. Inspired by the reviewer’s comments, we have done a number of experiment as well as corresponding finite element analysis for the temperature gradient in the system, which unambiguously proved that the signal we measured are NEITHER the local spin Seebeck effect NOR the anomalous Nernst effect.

MnPS₃ is highly electrically insulating. Thus there are only two possibilities that an in-plane magnetic field dependent second harmonic non-local signal can be detected in our device configuration: magnon transport and phonon transport (the later should be aided by local spin Seebeck/anomalous Nernst effect to be detected in our device configuration). Both effects would have second harmonic non-local signal that have a $\cos\theta$ dependence on the angle θ between the magnetic field and the direction perpendicular to the detector Pt electrode.

Nonetheless, compares to CrBr₃ where inside each layer the magnetic coupling is ferromagnetic, and YIG where there are net magnetization due to the ferrimagnetic nature of the material, MnPS₃ is has Ising type anti-ferromagnetic order within each Mn atomic layer (see Figure R2 below). Thus, it is much less likely that the Pt/MnPS₃ interface would be magnetized via proximity effect. This material properties of MnPS₃ makes us believe that the anomalous Nernst effect to be unlikely; while in the following we have three strong evidence that excluded anomalous Nernst effect or local spin Hall effect to affect our results.

Figure R2. The schematic of the interface of Pt/MnPS₃, Pt/CrBr₃ and Pt/YIG.

Here we provide three additional experimental evidences that excluded the possibility of local spin Seebeck effect as well as anomalous Nernst effect:

- a) **To clarify whether the heat is carried by phonon or by magnon**, we have fabricated a non-local device with a number of electrodes on MnPS₃. As depicted in Figure R3a below, we name four of the electrodes as Detector 1, Injector, oxidized Cu strip and Detector 2, respectively. All electrodes are made from Pt except for the oxidized Cu strip. The oxidized Cu strip is made from 10nm thick copper without any protection capping layer and then is exposed to ambient condition for oxidation. We have made sure that the Cu strip was conductive right after the deposition and not conductive after oxidation. The intention of the oxidation is to reduce the thermal conductivity of the copper strip to 4 W/m*K [A. Kusiak et al. Eur. Phys. J. Appl. Phys 35, 17-27(2006)], so that it is much lower than a Pt electrode in terms of thermal conductivity (72W/m*K for Pt). The oxidized copper strip merely acts as surface absorbates which only affect the top surface of MnPS₃ and would not act as a strong heat sink. In another word, the oxidized Cu strip should perturb magnon transport much more than phonon transport in MnPS₃, since the in-plane to out-of-plane ratio of magnetic coupling strength is 405:1 [JPCM 10, 6417-6428 (1998)] while the in-plane to out-of-plane ratio of thermal conductivity is only 6:1[ACS Nano 14, 2424-2435 (2020)]. The strong in-plane versus out-of-plane anisotropy in the magnetic exchange suggests that the Pt detectable magnon transport goes through only a few top layers of the sample, while the phonon transport generally goes through the whole layers of the sample. Being only the surface absorbates, the oxidized copper strip is expected to perturb the magnon transport dramatically, while the phonon transport remains robust against such perturbation.

An AC signal is applied through the Injector electrode, and the signal is measured simultaneously from Detector 1 and Detector 2. We found strong signal from Detector 1 (right next to the Injector) and no signal from Detector 2 (the oxidized copper strip is between Detector 2 and the Injector). Since the temperature gradient between Injector and Detector 2 should be finite as the case for Detector 1, which is confirmed by finite element analysis (see Figure R3c). The absence of the non-local inverse spin Hall signal from Detector 2 proves that phonon transport is not the cause of the non-local signal from the Detector electrode. We have added this data in the revised Supplementary Figure S9, with a short description.

Figure R3 (Figure S9). Non-local magnon signal detection with different MnPS₃ device geometries. (a) Schematic of non-local measurement on a MnPS₃ device with oxidized Cu strip. (b) Temperature dependence of $V_{2\omega}$ at $\theta = 0$ for the detector located at the left or right side of the oxidized Cu strip. (c) Finite element analysis of the temperature distribution in MnPS₃ device with oxidized Cu strip.

- b) **To quantify the effect of the anomalous Nernst effect in our experimental system**, we have measured the non-local second harmonic signal with an applied magnetic field of up to 14 T rotated in the x - z plane (see inset in Figure R4a below). Since there is finite temperature gradient along the x axis from our device configuration as shown from the finite element analysis, the temperature gradient along x also induces the Hall voltage along the y axis in the presence of the magnetization along the z axis (ANEx). According to PRB 101, 205407, ANEx and ANEz are of similar magnitude, where ANEz refers to the Hall voltage along y induced by the temperature gradient along z in the presence of the magnetization along the x axis (ANEz).

The angle of the magnetic field with respect to the z axis is marked as φ . An injection current of 100 μA is applied to the injector of our MnPS₃ device. We can see from Figure R4a that the data fits well to a $\sin\varphi$ function, in which the signal is zero when the magnetic field is along the z axis (perpendicular to the sample plane). From Figure R4a one can also see that only the magnetic field component along the x axis could produce non-zero non-local second harmonic signal. Figure R4b shows minimal magnetic-field dependence of the non-local signal with the magnetic field along the z axis (i.e., $\varphi=0$). This data

proves unambiguously the absence of anomalous Nernst effect with magnetic field perpendicular to the MnPS₃/Pt interface (ANEx), because the finite element calculation shows a finite temperature gradient along x . The absence of ANEx points to the absence of ANEz.

In fact, MnPS₃ is a layer antiferromagnet where the spin within one Mn atomic layer is aligned antiferromagnetically with a coupling constant J that amount to about 106T of magnetic field, which far exceeds the magnetic field applied in the experiment. It is natural that the MnPS₃/Pt interface remains non-magnetized. We have added this data in Supplementary Figure S10, with a short description.

Figure R4 (Figure S10). The absence of the anomalous Nernst effect in MnPS₃ magnon valve. (a) The non-local second harmonic signal as a function of angle ϕ between the external magnetic field ($B=9T$) and the z direction, angle ϕ is determined the same as shown in the inset.(b) The absence of magnetic field dependence of $V_{2\omega}$ at $\phi = 0$.

- c) **Distance dependent measurement on our MnPS₃ device** (see Figure R5 below). The decay length of the second harmonic thermal magnon signal is measured to be $\sim 3300\text{nm}$, which is consistent with previous report in the literature (e.g. $\sim 2800\text{ nm}$ for 16-nm MnPS₃, 1100 nm for 8-nm MnPS₃ as reported in PRX 9, 011026 (2019)). As shown in Fig. R5, this decay length is much longer than a decay length of the temperature gradient from the finite element calculation that represents how far the phonon carries the heat in space. **Thus, the longer decay length in the heater-detector distance dependence suggests that the signal cannot be explained by the phonon transport.** We have added the data in revised Supplementary Information S11.

Figure R5 (Figure S11). The heater-detector distance-dependent signal and temperature in MnPS₃ device. Left axis: The distance-dependent non-local second harmonic signal $V_{2\omega,0}$ and relaxation length fitting [Ref. 10] at 2K and 9T; Right axis: The finite element simulation of distance-dependent temperature of MnPS₃ under the same device configuration. The decay length of the second harmonic thermal magnon signal is measured to be about 3300nm.

Second, about the relative location of gate electrode. We would also like to stress that our theoretical model is based on the magnon band structure of MnPS₃ which provides the Seebeck coefficient for magnon transport in our device. Based on this model, the non-local voltage is a direct consequence of the spin current injected into the detector electrode given finite temperature gradient in the device provided by the injector electrode and the gate electrode. There is no restriction on the relative location of the gate and the injector in order to make our magnon valve work. The advantage of our model is that the details of the temperature gradient is absorbed into the three global parameters C , α and β , which are fixed for any particular device geometry operating at a certain base temperature and magnetic field. We found that in the injector-detector-gate configuration, the general behavior is similar (see Figure R6b). Finite element analysis shows that the variation of the temperature gradient is also similar for the gate located at the right or left side of the detector electrode (see Figure R6c&d), while the temperature of the MnPS₃ below the detector electrode is slightly higher for the case of the injector-detector-gate configuration, which may be the cause of the slight difference observed experimentally. We have added this data in Supplementary Figure S12, with a short explanation.

Figure R6 (Figure S12). Operation of a MnPS₃ magnon valve with different device geometries. (a) Schematics of non-local measurement on a MnPS₃ device with different gates. (b) $V_{2\omega,0}$ versus DC gate current I_{gate} at $B = 9\text{T}$ and temperature of 2K with different geometries. (c)(d) Finite element analysis of the temperature distribution in MnPS₃ device for the gate located at the left (c) or right (d) side of the detector electrode.

The parameters used in the finite element analysis are listed below:

Pt	Conductivity	8.9E6[S/m]	COMSOL Material database
	thermal conductivity	71.6[W/(m*K)]	COMSOL Material database
MnPS ₃	in-plane thermal conductivity	6.3[W/(m*K)]	ACS Nano,14, 2424–2435(2020)
	through-plane thermal conductivity	1.1[W/(m*K)]	ACS Nano,14, 2424–2435(2020)

SiO ₂	thermal conductivity	1.38[W/(m*K)]	CRC Handbook of Chemistry and Physics (92nd ed.).p12.213
Si	thermal conductivity	130[W/(m*K)]	COMSOL Material database
MnPS ₃ /SiO ₂	through-plane thermal resistance	5E-7[K*m ² /W] [#]	Computational Materials Science, 142, 1–6 (2018)
Pt/MnPS ₃	through-plane thermal resistance	1.4E-7[K*m ² /W] [§]	PHYSICAL REVIEW B 101, 205407 (2020)

Table R1. The parameters used in the finite element analysis. [#]There is no data found for MnPS₃/SiO₂ in the literature, we used value from through-plane thermal resistance between MoS₂/SiO₂ instead. [§]There is no data found for Pt/MnPS₃, we used estimated value for CrBr₃/Pt in the literature.

2) A similar sign reversal of the spin Seebeck signal with the injector heating current has been reported before for the 2D ferromagnet CrBr₃ [Phys. Rev. B 101, 205407 (2020)]. The shape of the signal modulation has a striking resemblance to the ones reported here, and can be explained by the simple heat profile under the detector contact and the inclusion of the anomalous Nernst effect. The authors should take this into account and provide a discussion along these lines, including/excluding the possible effects and making a proper comparison with previous experiments.

We would like to thank the reviewer for pointing out a potentially confusing point. It is a very good opportunity for us to clarify the three significant differences between our work and the work in PRB 101, 205407 (2020):

First of all, the channel materials are very different. MnPS₃ is a layered anti-ferromagnet with Ising-type anti-ferromagnetic coupling in the sample plane, while CrBr₃ is a 2D ferromagnet. This means that there is zero magnetic moment in each layer of MnPS₃, while each layer of CrBr₃ is magnetic. Such difference reflects strongly in the magnon spectra of the two materials [PRX 8, 011010 (2018), PRB 103, 024424 (2021)], and also reflects strongly in the existence of the magnetization at the Pt-CrBr₃ interface and the absence of which at the Pt-MnPS₃ interface, resulting in the observation of a large anomalous Nernst signal in CrBr₃ (PRB 101, 205407) and the absence of which in our work (see discussions in Reviewer #2, comment #1).

Second, the $R_{2\omega}$ vs. I_{in} curves are very different. We have plotted the $R_{2\omega}$ vs. I_{in} in Figure R7 (revised Supplementary Figure S13) for a couple of MnPS₃ devices below. In order to compare with the work on CrBr₃, the gate electrodes in our devices are floating during the measurement. It can be seen that the shape of the $R_{2\omega}$ vs. I_{in} of MnPS₃ device is very different from that of the CrBr₃ device shown in PRB 101, 205407. Interestingly, the shape of the $R_{2\omega}$ vs. I_{in} for CrBr₃ device has some

resemblance with the $V_{2\omega}$ vs. I_{in} for our MnPS₃ devices, which would be a good topic for future works.

Third, the devices are very different. Our work realized the first diffusive magnon valves in which a gate current controls whether the injected signal can be detected or not, which readily enables digital logic operation; PRB 101, 205407 (2020) describe a non-local response curve for the input signal without any external gate control.

According to the reviewer's comment, we have added a new section (S7) in the revised Supplementary Information to discuss the differences between our work and the work in PRB 101, 205407 (2020). The added discussion is colored in blue in the revised Supplementary Information.

Figure R7 (Figure S13). The $R_{2\omega}$ vs. I_{in} for different MnPS₃ devices with zero gate current.

3) The authors never mention the presence or absence of electrically-injected magnons, i.e. a first-harmonic signal. It should be at least mentioned that the authors did not see a first-harmonic signal above their noise floor, and give a value for that, so an upper bound for a possible signal is given.

We thank the reviewer for pointing out the need to mention the absence of electrically-injected magnons (first harmonic signal) in the beginning of the main text. As a matter of fact, we have written in page 10, line 237: “Furthermore, the simulation gives vanishingly small first harmonic response $V_{1\omega,0}$, which agrees with the physics of thermal magnon excitation and it is indeed what we observed experimentally (see Supplementary Fig. S6).” We have plotted the absence of the first-harmonic signal together with the finite second harmonic signal in the original Supplementary Figure S6. To improve the visibility of our discussion on electrically-injected magnons, we have added the following discussion at page 6, line 131 of the revised manuscript: “It is worth noting that magnons injected by exchange interactions are absent (i.e., there is zero first harmonic non-local signal $V_{1\omega}(\theta)$ with a π periodicity to the angle of the in-plane magnetic field) in our MnPS₃ devices (see Supplementary Figure S6), which is consistent with previous studies [PRX 9, 011026 (2019)].” The added discussion is highlighted in yellow in the revised manuscript.

4) The angular dependence of the signals shows that the spin Seebeck signal is proportional to the net magnetization induced by the field due to the canting of the magnetic moments. However, line 124 of page 5 gives the impression that this is due to a change of the antiferromagnetic magnon modes in MnPS₃, which should appear with the field applied along the Néel vector. The authors should clarify this point.

We thank the reviewer for bring up this point that might cause confusion. Indeed, when the applied magnetic field B is along the Néel vector, the energy of the magnon modes will be changing linearly with B ; when the applied B field is perpendicular to the Néel vector, the energy of the magnon modes will be changing quadratic with B . However, such linearity or quadratic relation between magnon energy and the magnetic field produces higher order effects in $V_{2\omega}$ as compares to the magnetization from the canting of the spins in the Mn layers. Specifically, from Eq. (3) at page 8, line 185 of the main text (page 9, line 200 in the revised main text), we can see that the linearity comes from the $\sin\psi$ term located at the beginning of Eq. (3). Here ψ is the canting angle of the spins from its easy axis at finite in-plane magnetic field. For small magnetic field, the canting angle is proportional to the in-plane magnetic field. We have modified our manuscript at page 5, line 126 to clarify this point: “This is consistent with the fact that the canting of the spins along the x direction is proportional to B , when B is small compares to the effective magnetic field of 106T for the exchange interactions between the nearest neighboring Mn atoms²⁸.”

Note that in the previous version of the manuscript, we used θ to denote the canting angle. For clarity, we have now changed the Greek letter for canting angle from θ to ψ

at page 9, line 200, line 203 and line 204 in the revised main text, due to the fact that we have already defined θ as the angle of the in-plane magnetic field with respect to the Pt electrode. The Greek letter representing the canting angle is also changed from θ to ψ in the revised Supplementary Information. All the revision and addition are highlighted in yellow in the main text and colored in blue in the Supplementary Information.

5) What is the sign of the signal shown here compared to previous results in MnPS₃ [ref. 26], YIG [ref. 10] and CrBr₃ [Phys. Rev. B 101, 205407 (2020)]? The positive and negative voltage probes should be clearly indicated in Fig. 1.

We thank the reviewer for bringing up the definition of the sign of the signal. We have been very careful about getting the right sign as well, which can be summarized below in Figure R8. Our definition of positive and negative voltage probes is completely the same as those in previous works on MnPS₃ [Ref. 26], on CrBr₃ [PRB 101, 205407 (2020)] and on YIG [Ref. 10] (for non-local signal with distance larger than ~250 nm in Ref. 10). According to the reviewer's comment, we have modified Fig. 1 by indicating the positive and negative voltage probes. Here we would like to mention that PRB 101, 205407 (2020) measured a "negative" spin Seebeck signal because the "positive" direction of the magnetic field they used is the opposite of ours (our definition of a positive magnetic field is shown in Figure R8), while the definition of positive and negative voltage probes are the same for both papers.

Figure R8. Schematic of non-local measurement on a typical MnPS₃ device via the low-frequency lock-in technique.

6) In order to extract a possible contribution from the anomalous Nernst effect the authors should also present an out-of-plane magnetic field angular dependence. This would also help clarifying the heat gradient directions in their system.

As discussed in the response to question #1 of reviewer #2, we have measured the second harmonic signal with vertical magnetic field of up to 14 T and an injection current of 100 μ A in our MnPS₃ device, and could not detect any non-local second harmonic signal, drastically different from the case of in-plane magnetic field

perpendicular to the Pt electrode. In conjunction with results in PRB 101, 205407 (2020), this proves the absence of anomalous Nernst effect in our MnPS₃ device. We have added this data in Supplementary Figure S10, with a short explanation.

Reviewer #3 (Remarks to the Author):

In this manuscript, the authors have claimed the realization of a magnon valve based on van der Waals anti-ferromagnetic insulator few layers MnPS₃. They show the tunability of the second harmonic non-local Voltage by applying a DC current through a metal “gate” located between the injection and detection electrodes. The behavior of the non-local signal is then simulated using the spin Seebeck coefficient based on a semi-classical Boltzmann transport theory with 3 free parameters.

Although the characteristic features of thermal magnon transport seem reasonably convincing I have several comments to be addressed by the authors:

We are glad that the reviewer found our result convincing and we appreciate the reviewer’s comments that help us substantially improve our manuscript.

1) It is quite common that magnetic vdW crystals are sensitive to ambient conditions and most of the time reactive to oxygen or humidity exposure. While some vdW materials such as most of the TMDs can resist to even rather high temperature (300degC), it has been shown that MnPS₃ would degrade. Do the authors have any way of confirming the integrity and quality of the topmost layers of their crystals? Would encapsulation be an option to avoid any damages?

Indeed a lot of vdW crystals are quite sensitive to ambient conditions. Many of the vdW magnetic materials, including but not limited to CrI₃, CrBr₃, Fe₃GeTe₂, etc., can degrade quickly when exposed to air. Similarly, the temperature stability is also an issue for a lot of vdW crystals. In the beginning of our experiment, we have carefully checked the air stability and thermal stability of MnPS₃. We found that few-layer MnPS₃ is quite air stable. We have taken the optical micrograph of few-layer MnPS₃ on SiO₂ substrate right after exfoliation and 8 months in air after exfoliation, which has no visible changes (see figure R9a below, also in the revised Supplementary Figure S14a). We found that few-layer MnPS₃ is also thermally stable up to 350 degC in air, which is well above the highest temperature (150 degC) during our device fabrication process. We have put few-layer MnPS₃ in a hot plate and have taken optical micrographs before heating and after heating to 350 degC in air (see figure R9b below, also in the revised Supplementary Figure S14b), where no visible changes are found. Thus, the stability of MnPS₃ is another advantage of the material as compares to many less stable layered magnetics materials.

Figure R9 (Figure S14). Stability test of few-layer MnPS₃. (a) the optical micrograph of few-layer MnPS₃ on SiO₂ substrate right after exfoliation and 8 months after exfoliation. (b) optical micrographs of few-layer MnPS₃ on SiO₂ substrate before heating and after heating to 150, 250 and 350°C for 10 minutes in air. (c) The device performance of our MnPS₃ magnon device right after fabrication and after 8 months.

That being said, optical micrographs can only tell us whether serious degradations happened. So we still paid special attention in handling the samples, and make sure that the sample does not unnecessarily expose to ambient conditions. We have

measured few-layer MnPS₃ devices right after fabrication and 8 months after fabrication, and did not see substantial degradation in the signal quality. We have put the data in the revised Supplementary Figure S14c and also in Figure R9c for your convenience. Encapsulation with BN is an effective method to improve the device stability for the highly reactive vdW crystals. Thus, using BN encapsulation should also provide MnPS₃ devices with more protection, but will at the same time increase the difficulty of a good contact between MnPS₃ and the Pt electrode. A very good two-dimensional contact is needed for the detector electrode to receive spin angular momentum injection from the non-equilibrium MnPS₃ magnons. Since the devices work well without encapsulation, we employ the precaution of avoiding prolonged exposure of the device to air and of storing the devices in Argon environment while they are not being measured or processed. This procedure works well for us in the experiment with MnPS₃.

2) The non-local signal is discussed as arising from the temperature gradient generated by the injection and the detection electrodes and modulated by the “gate” electrode. I am not sure to understand fully the microscopic mechanism of the “gating” electrode. Can the authors give a more detailed description? Could it be that the suppression of the non-local signal is simply due to an overheating of the junction with the gate electrode as observed at higher temperatures (above 20K)?

We thank the reviewer for bringing up this important point. In the beginning of our experiment, we have indeed been trying to attribute the tuning effect of the gate current to simply the thermal effect that heat up the whole crystal to temperature higher than 20K. However, the fact that: 1) there are two zero points for gate current I_0 and I_0' , and 2) between I_0 and I_0' , the second harmonic voltage changes sign as compares to gate current smaller than I_0 , cannot be explained by such picture. The microscopic mechanism of the “gating” electrode lies in the line shape of V_{ISHE} shown in Fig. S4 (a). We have added more discussion in page 9, line 212 to clarify the microscopic mechanism of the non-local signal: “Consequently, the thermally driven magnon spin current J_m will first increase, and then decrease with a general input current. In our real-time lock-in measurement, the first part will give a positive signal since more magnons are accumulating below the detector electrode with a non-zero input from the injection electrode. While in the second decreasing part, less magnons are accumulating below the detector electrode with applying the injection current, which equals to magnons flowing away from the detector electrode, resulting in a negative signal according to ISHE.” Also, from our finite element simulation, our sample temperature has never reached temperatures higher than 20K, which is consistent with our theoretical model discussed above.

3) I am not very convinced by the simulations obtained by fitting 3 free parameters. Although the discussion on the obtained values can have some credit in my point of view, I believe that some other external parameters could be used to refine or remove some parameters. One example could be to look at the “gate” dependence of the ratio

$V_{2\omega,in}/V_{2\omega,Gate}$ (as described in Supplementary) for various temperature and find a relation between alpha and beta.

The reviewer brings up an important point worth careful discussion: could we further reduce the parameters in the simulation. The discussion of these parameters can be found in Supplementary Section S5, we shall discuss them in more details here:

1) The overall parameter C in equation (4) mainly includes factors that relates a spin current injected into the detector Pt electrode to the voltage generated in the electrode; it also includes the dimensionless prefactor of the magnon relaxation time $1/\eta_{j,k}$ after summing contributions from all the magnon bands j and the momentum k in the first Brillouin Zone, which brings in various effects including crystal quality, temperature and magnetic field. Thus this parameter is difficult to be removed yet very easily determined via fitting to an experimental curve.

2) The parameter β characterizes the effectiveness for the injection and gating current to be converted to temperature gradient in the device. In principle it could be obtained from parameters such as the resistivity of the Pt strips, the thermal conductivity of the Pt strips and the Au electrodes that are connected to the Pt strips (the 200nm Pt strips are connected to Cr/Au electrodes which extended into bonding pads for the devices), the thermal conductivity of the MnPS₃ crystal, as well as thermal resistivity of all the material interfaces. Considering the complexity of all these external parameters and our goal to derive a predictive effective model, the parameter β is best to be determined via fitting to an experimental curve.

3) The parameter α is the ratio of the effective strength of the injector electrode over the gate electrode in terms of their influence to the detector-MnPS₃ interface. This ratio is also device dependent, mainly depending on the device structure, the channel length and thickness, as well as the material of the channel and the electrodes. If equation (4) is a linear function of the temperature gradient, parameter α could be the easiest to be obtained from additional tests, such as the one we did in Supplementary Information S5 (e.g., measure $V_{2\omega}$ for $I_{in} = 100\mu A$ and $I_{gate} = 0\mu A$ and $I_{in} = 0\mu A$ and $I_{gate,ac} = 100\mu A$). However, since S is an integral function that contains the temperature, and it is highly non-linear as shown in Supplementary Figure S4a, there is no simple relation between $V'_{2\omega,in}/V'_{2\omega,gate}$ and α . Thus, it is still the most straightforward to use three parameters in the simulation. We have extended the discussion in the revised Supplementary Information S5 to clarify this issue, and the revised discussion is colored in blue.

4) Other minor typos and comments:

- Line 57 vdw magnets have weak...
- Fig 3b on the y-axis it should be $V_{2\omega}$ and not $V_{2\omega,0}$

We thank the reviewer for pointing out the typos. We have corrected the typos in the revised manuscript. The corrections are highlighted in yellow in the revised main text.

- Line 152 I believe the Off states should be limited by the resolution of the measurement setup and not exactly 0 nV.

We agree with the reviewer that there is always a limit as to how accurately we could measure “0”; we would also like to stress that since the signal goes from positive to negative, the zero point (for the physical quantity itself, not the noise-limited measured values we could get) is guaranteed. To reflect the comments from the reviewer, we have modified page 7, line 169 in the revised manuscript to be: “In fact, Fig. 3c already illustrates the operation of a diffusive magnon based NOT gate, which shows finite output ($V_{2\omega} = 196$ nV) at zero input ($I_{\text{gate}} = 0$) and zero output ($V_{2\omega} = 0$), here “0” means below the noise floor of our measurement system, which is < 1 nV) at finite input ($I_{\text{gate}} = 150\mu\text{A}$).”

Reviewer #1 (Remarks to the Author):

The authors have satisfactorily addressed all issues I raised in my last report, and I think they also reasonably clarified the questions of other reviewers through extended experiments and simulations. So I recommend the publication of current manuscript in Nature Communications.

We are glad that the reviewer considered our revision satisfactory and recommend the publication of our current manuscript in Nature Communications. We highly appreciate the helpful comments from the reviewer that helped us greatly improve our manuscript.

Reviewer #2 (Remarks to the Author):

In their response, updated manuscript, and supplementary information the authors bring up new data and analysis which address the questions and concerns raised by me and the other two reviewers. My main previous concern was that the magnon transport and modulation shown is solely of local origin (i.e. local spin Seebeck effect without magnon transport). With the new experiments, simulations and arguments, I am convinced that this is likely not the case. Indeed, the authors do a very careful job addressing my comments, which I appreciate. I find the new version of the manuscript much more thorough and attractive to researchers in magnonics and magnetic 2D materials. Therefore, I believe this manuscript can now be published in Nature Communications.

We are grateful for the questions raised by the reviewer, which prompted us to substantially improve our manuscript through additional experiment, simulation and analysis. Our point-by-point reply to the additional comments of the reviewer is listed below. All revisions in this round of review process are marked in blue in the revised manuscript and supplementary information.

Nevertheless, before publication I would still like to ask the authors to address the following points:

- 1) The discussion brought up in the response letter on my main point (local versus non-local magnon detection) is overlooked in the new version of the supplementary material. Even though the figures are added, there is no text accompanying it. I understand that the response letters are also published, but I would strongly suggest the authors to add the discussion accompanying figures R3 to R6 and table R1 to the supplementary information.

We agree with the reviewer that the supplementary information should contain more text explanation to facilitate the understanding of our work by the readers, as people

might not regularly look into the response letters. We have added the following explanations:

1) In Supplementary Information Section 9:

“Section S9 to S12 contain additional experimental evidences that excluded the possibility of local spin Seebeck effect as well as anomalous Nernst effect in the MnPS_3 magnon valves.

First of all, to clarify whether the heat is carried by phonon or by magnon, we have fabricated a non-local device with a number of electrodes on MnPS_3 . As depicted in Fig. S9 below, we name four of the electrodes as Detector 1, Injector, oxidized Cu strip and Detector 2, respectively. All electrodes are made from Pt except for the oxidized Cu strip. The oxidized Cu strip is made from 10nm thick copper without any protection capping layer and then is exposed to ambient condition for oxidation. We have made sure that the Cu strip was conductive right after the deposition and not conductive after oxidation. The intention of the oxidation is to reduce the thermal conductivity of the copper strip to 4 $\text{W/m}\cdot\text{K}$ [10], so that it is much lower than a Pt electrode in terms of thermal conductivity (72 $\text{W/m}\cdot\text{K}$ for Pt). The oxidized copper strip merely acts as surface absorbates which only affect the top surface of MnPS_3 and would not act as a strong heat sink. In another word, the oxidized Cu strip should perturb magnon transport much more than phonon transport in MnPS_3 , since the in-plane to out-of-plane ratio of magnetic coupling strength is 405:1 [1] while the in-plane to out-of-plane ratio of thermal conductivity is only 6:1 [11]. The strong in-plane versus out-of-plane anisotropy in the magnetic exchange suggests that the Pt detectable magnon transport goes through only a few top layers of the sample, while the phonon transport generally goes through the whole layers of the sample. Being only the surface absorbates, the oxidized copper strip is expected to perturb the magnon transport dramatically, while the phonon transport remains robust against such perturbation.

An AC signal is applied through the Injector electrode, and the signal is measured simultaneously from Detector 1 and Detector 2. We found strong signal from Detector 1 (right next to the Injector) and no signal from Detector 2 (the oxidized copper strip is between Detector 2 and the Injector). Since the temperature gradient between Injector and Detector 2 should be finite as the case for Detector 1, which is confirmed by finite element analysis (see Fig. S9c). The absence of the non-local inverse spin Hall signal from Detector 2 proves that phonon transport is not the cause of the non-local signal from the Detector electrode.”

2) In Supplementary Information Section 10:

“To quantify the effect of the anomalous Nernst effect in our experimental system, we have measured the non-local second harmonic signal with an applied magnetic field of up to 14 T rotated in the x - z plane (see inset in Fig. S10 below). Since there is finite temperature gradient along the x axis from our device configuration as shown from the finite element analysis, the temperature gradient along x also induces the Hall voltage along the y axis in the presence of the magnetization

along the z axis (ANEx). It's considered that ANEx and ANEz are of similar magnitude¹², where ANEz refers to the Hall voltage along y induced by the temperature gradient along z in the presence of the magnetization along the x axis. The angle of the magnetic field with respect to the z axis is marked as φ . An injection current of 100 μA is applied to the injector of our MnPS₃ device. We can see from Fig. S10a that the data fits well to a $\sin\varphi$ function, in which the signal is zero when the magnetic field is along the z axis (perpendicular to the sample plane). From Fig. S10a one can also see that only the magnetic field component along the x axis could produce non-zero non-local second harmonic signal. Figure S10b shows minimal magnetic-field dependence of the non-local signal with the magnetic field along the z axis (i.e., $\varphi=0$). This data proves unambiguously the absence of anomalous Nernst effect with magnetic field perpendicular to the MnPS₃/Pt interface (ANEx), because the finite element calculation shows a finite temperature gradient along x . The absence of ANEx points to the absence of ANEz¹².

In fact, MnPS₃ is a layer antiferromagnet where the spin within one Mn atomic layer is aligned antiferromagnetically with a coupling constant J that amount to about 106T of magnetic field, which far exceeds the magnetic field applied in the experiment. It is natural that the MnPS₃/Pt interface remains non-magnetized. We have added this data in Supplementary Figure S10, with a short description.”

3) In Supplementary Information Section 11:

“We have measured the distance dependence of $V_{2\omega,0}$ (black dots in Fig. S11) and the experimental data is fitted to $V_{2\omega} = \frac{C_0}{\lambda} * \frac{\exp(d/\lambda)}{1-\exp(2d/\lambda)}$ [13], where C_0 is a factor characterizing the magnitude of the second harmonic signal, λ is the decay length of the diffusive magnons. The fitting gives $C_0 = -2.4\pm 0.2 \mu\text{V}$ and $\lambda = 3300\pm 200$ nm, which is consistent with previous report in the literature (e.g. $\lambda\sim 2800$ nm for 16-nm MnPS₃, 1100 nm for 8-nm MnPS₃ [14]). As shown in Fig. S11, this decay length (red curve) is much longer than a decay length of the temperature gradient from the finite element calculation (blue curve) that represents how far the phonon carries the heat in space. Thus, the longer decay length in the heater-detector distance dependence suggests that the signal cannot be explained by the phonon transport.”

4) In Supplementary Information Section 12:

“Two different device geometries, namely, the injector-gate-detector configuration as well as the injector-detector -gate configuration is tested. We found that in the injector-detector-gate configuration, the general behavior is similar (see Fig. S12b). Finite element analysis shows that the variation of the temperature gradient is also similar for the gate located at the right or left side of the detector electrode (see Fig. S12c&d), while the temperature of the MnPS₃ below the detector electrode is slightly higher for the case of the injector-detector-gate configuration,

which may be the cause of the slight difference in $V_{2\omega}$ vs. I_{gate} observed experimentally.”

5) Supplementary Table S1 is added into the supplementary information after the text shown in 4) above.

2) I would also suggest the authors to include the fitting results of Fig. R5 (S11) with their respective errors.

We have added discussion in Supplementary Information S11: “We have measured the distance dependence of $V_{2\omega,0}$ (black dots in Fig. S11) and the experimental data is fitted to $V_{2\omega} = \frac{C_0}{\lambda} * \frac{\exp(d/\lambda)}{1-\exp(2d/\lambda)}$ [13], where C_0 is a factor characterizing the magnitude of the second harmonic signal, λ is the decay length of the diffusive magnons. The fitting gives $C_0 = -2.4 \pm 0.2 \mu\text{V}$ and $\lambda = 3300 \pm 200 \text{ nm}$, which is consistent with previous report in the literature (e.g. $\lambda \sim 2800 \text{ nm}$ for 16-nm MnPS₃, 1100 nm for 8-nm MnPS₃ [14]). As shown in Fig. S11, this decay length (red curve) is much longer than a decay length of the temperature gradient from the finite element calculation (blue curve) that represents how far the phonon carries the heat in space. Thus, the longer decay length in the heater-detector distance dependence suggests that the signal cannot be explained by the phonon transport.”

3) In the response and new version of the SI, the authors report that their $V_{2\omega}$ versus I_{in} curves look similar to the $R_{2\omega}$ vs I_{in} for CrBr₃ [ref 27], but the curves reported for $R_{2\omega}$ vs I_{in} (Fig. R7 / S13) do not show a sign reversal of $R_{2\omega}$ as in Ref 27. Here I am assuming that $R_{2\omega} = V_{2\omega}/(I_{\text{in}})^2$ as it is conventionally used in literature. Perhaps the authors mean $R_{2\omega}$ vs I_{in} of Ref. 27 compared to their $V_{2\omega}$ vs I_{gate} ?

We would like to thank the careful reviewing from the reviewer. We realized that it was a typo in our previous response to the reviewer’s question in the first round of review. The sentence: “Interestingly, the shape of the $R_{2\omega}$ vs. I_{in} for CrBr₃ device has some resemblance with the $V_{2\omega}$ vs. I_{in} for our MnPS₃ devices...” should actually be: “Interestingly, the shape of the $R_{2\omega}$ vs. I_{in} for CrBr₃ device has some resemblance with the $V_{2\omega}$ vs. I_{gate} for our MnPS₃ devices...” We apologize for the typo in our previous reply.

Indeed, we use the convention $R_{2\omega} = V_{2\omega}/(I_{\text{in}})^2$ to obtain Fig. S13, thus $R_{2\omega}$ and $V_{2\omega}$ always have the same sign. In order to illustrate this matter better, we added Fig. S14 in the supplementary information showing $V_{2\omega}$ vs. I_{in} for the same set of MnPS₃

devices as shown in Fig. S13. The new Fig. S14 is also reproduced below for the reviewer's convenience. As can be seen in Fig. S14, $V_{2\omega}$ vs. I_{in} for MnPS₃ devices are very different from $V_{2\omega}$ vs. I_{in} for CrBr₃ devices [ref 27].

Fig. S14. The $V_{2\omega}$ vs. I_{in} for the same set of MnPS₃ devices with zero gate current as shown in Fig. S13.

Reviewer #3 (Remarks to the Author):

The authors have answered most of my remarks satisfactorily. However, I am still not convinced about the bold statement of the authors “strongly” suggesting that their simulation “captures the main physics behind the switching behavior of the MnPS₃ magnon valves” based on a fitting with 3 free parameters which seems in rather poor agreement at low I_{gate} ($I_{\text{gate}} < 50 \mu\text{A}$). The authors have brought good argument concerning why the different parameters are indeed complex to be obtained from an experimental point of view but the validity of a model should not depend on such consideration. In my opinion, I think that the authors should lower their claims concerning the simulation.

We agree with the reviewer that some deviation between experiment and simulation is present for low I_{gate} , which means that additional factors are still out there beyond our current model. Part of our model is phenomenological (i.e. Eq. (4)), which certainly warrants further refinement and research in future studies. We are glad that the reviewer brought up this point, and we have made modifications accordingly, in page 11, line 240 of the revised manuscript: “The above agreement between the simulation and the experimental data suggests that our model captures the physical trends behind the switching behavior of the MnPS₃ magnon valves.” We have also added discussion in page 11, line 250 of the revised manuscript: “It is also worth noting that, albeit good overall agreement is found between the experimental data and theoretical simulation with only three global parameters, appreciable difference between the experiment and simulation is found for small I_{gate} (i.e., $I_{\text{gate}} < 50 \mu\text{A}$), which hints the existence of additional factors and warrants further study.” All revisions in this round of review process are marked in yellow in the revised manuscript.

Point-by-point response:

Reviewer #2 (Remarks to the Author):

The authors have successfully addressed my comments from my previous report. I can now recommend the publication of this manuscript as is.

Response: Thanks a lot for the brief comment from reviewer 2. We have kept the manuscript as is with no changes, except for the response to the editorial requests.